



# Modelling the future evolution of glaciers in the European Alps under the EURO-CORDEX RCM ensemble

Harry Zekollari[1,2], Matthias Huss[1,3], Daniel Farinotti[1,2]

[1]Laboratory of Hydraulics, Hydrology and Glaciology (VAW), ETH Zürich, Zürich, Switzerland
[2]Swiss Federal Institute for Forest, Snow and Landscape Research (WSL), Birmensdorf, Switzerland
[3]Department of Geosciences, University of Fribourg, Fribourg, Switzerland

*Correspondence to*: Harry Zekollari (zharry@ethz.ch)

**Abstract.** Glaciers in the European Alps play an important role in the hydrological cycle, act as a source for hydroelectricity and have a large touristic importance. The future evolution of these glaciers is driven by surface mass balance and ice flow

processes, which the latter is to date not included in regional glacier projections for the Alps. Here, we model the future evolution of glaciers in the European Alps with GloGEMflow, an extended version of the Global Glacier Evolution Model (GloGEM), in which both surface mass balance and ice flow are explicitly accounted for. The mass balance model is calibrated with glacier-specific geodetic mass balances, and forced with high-resolution regional climate model (RCM) simulations from the EURO-CORDEX ensemble. The evolution of the total glacier volume in the coming decades is

relatively similar under the various representative concentrations pathways (RCP2.6, 4.5 and 8.5), with volume losses of about 47-52% in 2050 with respect to 2017. We find that under RCP2.6, the ice loss in the second part of the 21[st] century is relatively limited and that about one-third (36.8% ± 11.1%) of the present-day (2017) ice volume will still present in 2100. Under a strong warming (RCP8.5) the future evolution of the glaciers is dictated by a substantial increase in surface melt, and glaciers are projected to largely disappear by 2100 (94.4±4.4% volume loss vs. 2017). For a given RCP, differences in

future changes are mainly determined by the driving global climate model, rather than by the RCM that is coupled to it, and these differences are larger than those arising from various model parameters. We find that under a limited warming, the inclusion of ice dynamics reduces the projected mass loss and that this effect increases with the glacier elevation range, implying that the inclusion of ice dynamics is likely to be important for global glacier evolution projections.



## 1 Introduction

In the coming century, glaciers are projected to lose a substantial part of their volume, maintaining their position as one of the main contributors to sea-level rise (Slangen et al., 2017; Bamber et al., 2018; Marzeion et al., 2018; Moon et al., 2018;

Parkes and Marzeion, 2018). In the European Alps the retreat of glaciers will have a large impact, as glaciers play an important role for river runoff (e.g. Hanzer et al., 2018; Huss and Hock, 2018), hydroelectricity production (e.g. Milner et al., 2017; Patro et al., 2018) and for touristic purposes (e.g. Fischer et al., 2011; Welling et al., 2015; Stewart et al., 2016).

In order to understand how the ca. 3500 glaciers of the European Alps (Pfeffer et al., 2014) (**Fig. 1**a) will react to changing

$21^{st}$ century climatic conditions (e.g. Gobiet et al., 2014; Christidis et al., 2015; Frei et al., 2018; Stoffel and Corona, 2018), to date, models of various complexity have been used. Regional glacier evolution studies in the Alps (Zemp et al., 2006; Huss, 2012; Salzmann et al., 2012; Linsbauer et al., 2013) have focused on methods in which ice dynamics are not explicitly accounted for and the evolution of the glacier is based on parameterisations. One of the first studies to estimate the future evolution of all glaciers in the European Alps, was performed by Zemp et al. (2006), who used a statistical calibrated

equilibrium-line altitude (ELA) model to estimate future area losses. In another study, Huss (2012) modelled the future evolution of about 50 large glaciers with a retreat parameterisation, and extrapolated these findings to the entire European Alps using various methods. The future evolution of glaciers in the European Alps was also modelled as a part of global studies, relying on methods that parameterise glacier changes through volume/length-area scaling (Marzeion et al., 2012; Radić et al., 2014) and methods in which geometry changes are imposed based on observed changes (Huss and Hock, 2015).

These regional and global studies generally suggest a glacier volume loss of about 65-80% between the early $21^{st}$ century and 2100 under a moderate warming, and an almost complete disappearance of glaciers under an extreme warming scenario.

For certain well-studied Alpine glaciers, 3-D high-resolution ice flow models have been used to simulate their future evolution (e.g. Le Meur and Vincent, 2003; Le Meur et al., 2004, 2007, Jouvet et al., 2009, 2011; Zekollari et al., 2014).

These studies are of large interest to better understand the glacier dynamics and the driving mechanisms behind their future evolution, but individual glacier characteristics hamper extrapolations of these findings to the regional scale (Beniston et al., 2018). Given the computational expenses related to running such complex models, and due to the lack of field measurements needed for model calibration and validation (e.g. mass balance, ice thickness and surface velocity measurements), these models cannot be applied for every individual glacier in the European Alps. A possible alternative consists of using a

regional glaciation model (RGM), in which a surface mass balance (SMB) component and an ice dynamics component are coupled and applied over an entire mountain range, i.e. not for every glacier individually (Clarke et al., 2015). However, running a RGM at a high spatial resolution remains computationally expensive, and the discrepancy between the model complexity and the uncertainty in the various boundary conditions persists. Furthermore, Clarke et al. (2015) showed in a RGM study for western Canada that large present-day differences between observed and modelled glacier geometries can





locally exist after a transient simulation. For relative area and volume changes this was found to not be problematic, as they are relatively unaffected by these discrepancies (Clarke et al., 2015). However, for the European Alps, with many small glaciers that are largely controlled by topography and local effects, this discrepancy is expected to have a larger effect and impede a more detailed analysis of future individual glacier changes. To date, the most adequate and sophisticated method to

model the evolution of all glaciers in the European Alps thus consists of modelling every glacier individually with a coupled ice flow – surface mass balance model. A pilot study in this direction was recently undertaken by Maussion et al. (2018), with the newly released Open Global Glacier Model (OGGM), in which steady-state simulations are performed for every glacier worldwide  based on standard (i.e. non-calibrated) model parameters under the 1985-2015 climate.

Here, we explore the potential of using a coupled SMB – ice flow model for regional projections, by modelling the future evolution of glaciers in the European Alps with such a model. For this, we extend the Global Glacier Evolution Model (GloGEM) of Huss and Hock (2015) by introducing an ice flow component. We refer to this model as GloGEMflow in the following. Our approach is furthermore novel, as glacier-specific geodetic mass balance estimates are used for model calibration and the future glacier evolution relies directly on regional climate change projections from the EURO-CORDEX

(COordinated Regional climate Downscaling EXperiment applied over Europe) ensemble (Jacob et al., 2014; Kotlarski et al., 2014). This is, to our knowledge, the first regional glacier modelling study in the Alps directly making use of this high-resolution regional climate model (RCMs) output. In contrast to a forcing with a general circulation model (GCM), a RCM coupled to a GCM can provide information on much smaller scales, supporting a more in-depth impact assessment and providing projections with much detail and more accurate representation of localised events.

Through novel approaches in terms of (i) climate forcing, (ii) inclusion of ice dynamics, (iii) the use of glacier-specific geodetic mass balance estimates for model calibration, and by relying on a vast and diverse dataset on ground-truth data for model calibration and validation, we aim at reducing the considerable uncertainties in projections of future glacier evolution in the European Alps (Beniston et al., 2018).

## 2    Data

### 2.1 Glacier geometry

For every individual glacier in the European Alps, the outlines are taken from the Randolph Glacier Inventory (RGI v6.0) (RGI Consortium, 2017) (**Fig. 1**). These glacier outlines are mostly from an inventory derived by Paul et al. (2011) using Landsat Thematic Mapper scenes from August and September 2003 (for 96.7% of all glaciers included in the RGI). The surface hypsometry is derived from the Shuttle Radar Topography Mission (SRTM) version 4 (Jarvis et al., 2008) Digital Elevation Model (DEM) from 2000 (RGI Consortium, 2017).





The ice thicknesses of the individual glaciers is calculated according to the method of Huss and Farinotti (2012) in 10-m elevation bands, which relies on ice volume flux estimates. The horizontal distance ($\Delta x$) between the elevation bands is determined from the elevation difference ($\Delta y$) and the local surface slope ($s$):

$$\Delta x = \frac{\Delta y}{\tan s} \quad .$$
(1)

Subsequently, the glacier geometry is interpolated to an along flow regular horizontal grid. Through this approach, possible glacier branches and tributaries are not explicitly accounted for, avoiding complications and potential problems related to solving the mass transfer in these little-known connections. Glacier cross-sections are parameterised through a volume- and area-conserving approach in which they are represented as symmetrical trapezoids. These symmetrical trapezoids deviate from a rectangular cross-section by an angle $\alpha$ (see suppl. mat. Fig. S1). A value 45° is taken for $\alpha$, the effect of which is

assessed in the uncertainty analysis (section 6.3).

## 2.2 Climatic data

2-m air temperature and precipitation are used to represent the climatic conditions at the glacier surface for SMB calculations

(section 3.1). For the past (1951-2017), we rely on the ENSEMBLES daily gridded observational dataset (E-OBS v.17.0) on a 0.22° grid (Haylock et al., 2008). We prefer using an observational dataset compared to a re-analysis product, in order to ensure a close representation of past temperature and precipitation and certain events (e.g. the heat wave of the summer of 2003 (Beniston, 2004) , cf. **Fig. 2**b), allowing for detailed comparisons between observed and modelled surface mass balances on short time scales (section 4.1). Furthermore, the observational product has a higher resolution than the original

reanalysis data (ERA-INTERIM) and goes back further in time. Additionally, relying on higher-resolution RCM simulations forced with reanalysis data is not possible, as for several chains used for the future simulations (see next paragraph), a related simulation is not available. This would furthermore complicate the model validation for the past, as the past climatic conditions would be different for every model chain, while the observational data provides a single past temperature and precipitation forcing.

For the future, we use climate change projections from the EURO-CORDEX ensemble (Jacob et al., 2014; Kotlarski et al., 2014), from which all available simulations at 0.11° resolution (ca. 12 km horizontal resolution) are considered. This corresponds to a total of 51 chains, consisting of different combinations of nine RCMs, six GCMs and various realisations (r1i1p1, r12i1p1, r2i1p1, r3i1p1), forced with three representative concentration pathways (RCPs; **Fig. 2** and suppl. mat.

Table S1) (van Vuuren et al., 2011; IPCC, 2013). The three considered RCPs are (i) a peak decline scenario with a rapid stabilisation of atmospheric $CO_2$ levels (RCP2.6), (ii) a medium mitigation scenario (RCP4.5) and (iii) a high-emission scenario (RCP8.5). Notice that country-specific projections exist, such as the recently released CH2018 scenarios for





Switzerland (CH2018, 2018), which rely on simulations from the EURO-CORDEX ensemble, but these cannot be applied in a uniform way over the entire Alps.

To ensure a consistency between the observational (E-OBS, used for past) and RCM (EURO-CORDEX, used for future)
climatic data, a debiasing procedure is applied (Huss and Hock, 2015). Here, additive (temperature) and multiplicative (precipitation) monthly biases are calculated to ensure a consistency in the magnitude of the signal over the common time period. These biases are assumed to be constant in time and are superimposed on the RCM series. Furthermore, RCM temperature series are adjusted to account for differences in year-to-year variability between the observational and the RCM time series. Accounting for the differences in interannual variability is crucial to ensure the validity of the calibrated model
parameters for the future RCM projections (Hock, 2003; Farinotti, 2013). For each month $m$, the standard deviation of temperatures over the common time period is calculated for both the observational ($\sigma_{obs,m}$) and the RCM data ($\sigma_{RCM,m}$). For each month $m$ and year $y$ of the projection period, the interannual variability of the RCM air temperatures $T_{m,y}$ is corrected as:

$$T_{m,y,\text{corrected}} = \overline{T_{m,25}} + \left(T_{m,y} - \overline{T_{m,25}}\right)\phi_m \quad . \tag{2}$$

Here $\overline{T_{m,25}}$ is the average temperature in a 25-year period centered around $y$, and $\phi_m$ corresponds to $\sigma_{obs,m}/\sigma_{RCM,m}$. This
procedure ensures consistency in interannual variability, while allowing for future changes in the temperature variability given by the RCMs (**Fig. 2**).

**2.3 Mass balance**

The SMB model component is calibrated (section 3.1) with glacier-specific geodetic mass balances taken from the World Glacier Monitoring Service (WMGS) database (WGMS, 2018). Most of these geodetic mass balances were derived by Fischer (2011) (Austria), Berthier et al. (2014) (France, Italy and Switzerland) and Fischer et al. (2015b) (Switzerland). Many Alpine glaciers have a glacier-specific geodetic mass balance observation. For glaciers for which several geodetic SMB observations are available, the one closest to the reference period 1981-2010 is selected for model calibration. In case
no geodetic mass balance observation for the specific glacier is available, an observation from a nearby glacier is chosen, based on a combined criterion weighting both horizontal distance and the difference in area.

For model validation (section 4.1), we rely on in situ SMB observations provided by the WGMS Fluctuations of Glaciers Database (WGMS, 2018), consisting of 1672 glacier-wide annual balances and 12'097 annual balances for specific glacier
elevation bands.





## 3    Methods

GloGEMflow consists of a surface mass balance component (section 3.1), which is taken from GloGEM (Huss and Hock, 2015), and an ice flow component (section 3.2). These two components are combined to calculate the temporal evolution of

the glacier (section 3.3).

### 3.1 Surface mass balance modelling

Here, we briefly describe the SMB model component, with an emphasis on the settings specific to this study. For a more

elaborate description, we refer to the description in Huss and Hock (2015).

The model is forced with monthly temperature and precipitation series (section 2.2) from the E-OBS (past) or RCM (future) grid cell closest to the specific glacier. Accumulation is computed from precipitation and a threshold is used to differentiate between liquid and solid precipitation. This threshold is defined as an interval from 0.5°C to 2.5°C, within which the

snow/rain ratio is linearly interpolated. The melt is calculated for every grid cell from a classic temperature-index model (Hock, 2003), in which a distinction between snow, firn and ice melt is made based on different degree-day factors. Refreezing of rain and melt water is also accounted for and calculated from snow and firn temperatures based on heat conduction (see Huss and Hock, 2015). Huss and Hock (2015) showed that the added value of using a simplified energy balance model (Oerlemans, 2001) was limited, and that it did not perform better than the temperature-index model when

validated against SMB measurements.

For every individual glacier the climatic data is scaled from the gridded product to the individual glacier at a rate of 2.5% per 100 m elevation change for precipitation and by relying on monthly temperature lapse rates derived from the RCMs. Subsequently, model calibration parameters (degree-day factors, precipitation scaling factor) are adapted as a part of a

glacier-specific three-step calibration procedure that aims at reproducing the observed geodetic mass balance. In the first step, overall precipitation is multiplied with a scaling factor varying between 0.8 and 2.0. This initial step focuses on the precipitation, as this is the variable that is expected to be the most poorly reproduced due to resolution issues, spatial variability and local effects (e.g. Jarosch et al., 2012; Hannesdóttir et al., 2015; Huss and Hock, 2015). If this step does not allow reproducing the observed geodetic SMB within a 10% bound, in a second step the degree-day factors for snow and ice

are modified. Here, the degree-day factor of snow is allowed to vary between 1.75 and 4.5 mm d$^{-1}$ K$^{-1}$ and the degree-day factor of ice is adjusted to ensure a 2:1 ratio between both degree-day factors. In an eventual third and final step, the air temperature is modified through a systematic shift over the entire glacier (see Figure 2a in Huss and Hock, 2015 for more details).





### 3.2 Ice flow modelling

The ice flow is modelled explicitly with a flowline model for all glaciers longer than 1 km at the RGI inventory date. These 795 glaciers represent ca. 95% of the total volume and 86% of the total area of all glaciers in the European Alps (**Fig. 1** inset). For glaciers shorter than 1 km, mass transfer within the glacier is limited, and the time evolution is modelled through the $\Delta h$-parameterisation (Huss et al., 2010b), in line with the original GloGEM setup (Huss and Hock, 2015).

The dynamics of the ice flow component are solved through the Shallow-Ice Approximation (SIA) (Hutter, 1983), in which basal shear ($\tau$) is proportional to the local ice thickness ($H$) and the surface slope ($\frac{\partial s}{\partial x}$):

$$\tau = -\rho g H \frac{\partial s}{\partial x} \tag{3}$$

$g$=9.81 m s$^{-2}$ is gravitational acceleration, while the ice density $\rho$ is set to 900 kg m$^{-3}$. In our model, mass transport is expressed through a Glen (1955) type of flow law, in which the depth-averaged velocity $\bar{u}$ (m yr$^{-1}$) is defined as:

$$\bar{u} = \frac{2A}{n+2} \tau^n H. \tag{4}$$

Here $n = 3$ is Glen's flow law exponent, and $A$ is the deformation-sliding factor (Pa$^{-3}$ yr$^{-1}$) that accounts for the effects of the ice rheology on its deformation, sliding and various others effects (e.g. lateral drag). Basal sliding is implicitly accounted for through this approach and not treated separately from internal deformation, given the relatively large uncertainties associated with it. Basal sliding and internal deformation are both linked to the surface slope and the local ice thickness and have been shown to have similar spatial patterns on Alpine glaciers (e.g. Zekollari et al., 2013), justifying an approach in which both are combined (e.g. Gudmundsson, 1999; Clarke et al., 2015).

### 3.3 Time evolution and initialisation

The glacier geometry is updated at every time step through the continuity equation:

$$\frac{\partial H}{\partial t} = \nabla \cdot f + b \quad , \tag{5}$$

where $b$ is the surface mass balance (m w.e. yr$^{-1}$) and $\nabla \cdot f$ is the local divergence of the ice flux ($f = D\frac{\partial s}{\partial x}$). For a transect with a trapezoidal shape, with a basal width $w_b$ and a surface width $w_s$ (suppl. mat. Fig. S1), this becomes (cf. Oerlemans, 1997):

$$\frac{\partial H}{\partial t} = -\frac{1}{w_s} \left[ \left( \frac{w_b + w_s}{2} \right) \frac{\partial \left( D\frac{\partial h}{\partial x} \right)}{\partial x} + \left( D\frac{\partial h}{\partial x} \right) \frac{\partial \left( \frac{w_b + w_s}{2} \right)}{\partial x} \right] + b \quad , \tag{6}$$

where D is the diffusivity factor (m$^2$ yr$^{-1}$):

$$D = A \left( \frac{2}{n+2} \tau^n H^2 \right) \left( \frac{\partial s}{\partial x} \right)^{-1}. \tag{7}$$



The initialisation consists of closely reproducing the glacier geometry at the inventory date. At first, a constant climate is imposed, until a steady state is created, which represents the glacier in 1990 (**Fig. 3**). This constant climate corresponds to the 1961-1990 climate, to which a SMB perturbation is applied (detailed below). Subsequently, the glacier is forced with E-OBS data, and evolves transiently from 1990 until the glacier-specific inventory date (typically 2003). The glacier volume and length at the inventory date are matched by modifying two calibration variables (**Fig. 3**):

1. The deformation-sliding factor $A$, the calibration of which mainly determines the volume of the glacier at the inventory date.

2. A SMB offset in the 1961-1990 climatic conditions used to construct a 1990 steady-state glacier, which mainly determines the length of the glacier at the inventory date. Notice that through this approach, the glacier is not assumed to be in steady state at any point in time, but that an artificially modelled steady state is obtained by imposing a MB offset.

An optimisation procedure is used, in which at each iteration the deformation-sliding factor and the SMB offset are informed from previous iterations (see Appendix A for details). This results in a fast convergence to the desired state, i.e. a glacier with the same length and volume as the reference geometry (from Huss and Farinotti, 2012) at the inventory date. It should be noticed that the reference volume is itself a model result (Huss and Farinotti, 2012) and thus also holds uncertainties. The choice for the 1990 steady state and the effect this has on the calibration procedure is assessed in the uncertainty analysis (section 6.3).

## 4    Model validation

### 4.1 Surface mass balance

The modelled SMB is evaluated by using independent, observed glacier-wide annual balances and annual balances for glacier elevation bands (**Fig. 4**). As the aim is to evaluate the performance of the SMB model in this section, rather than the coupled SMB – ice flow model, these calculations are based on the glacier geometry at the inventory date. The observed glacier-wide annual mass balances are in general well reproduced, with a root-mean-square error (RMSE) of 0.71 m w.e. yr$^{-1}$, a Median Absolute Deviation (MAD) of 0.64 m w.e. yr$^{-1}$ and a systematic error (mean misfit) of -0.05 m w.e. yr$^{-1}$ (**Fig. 4**a,b). Furthermore, the good agreement between observed and modelled balances for glacier elevation bands ($r^2 = 0.63$; **Fig. 4**b,d) suggests that, despite not being calibrated to this, the SMB model distributes the annual SMB relatively well over elevation.



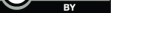

### 4.2 Glacier geometry

The glaciers are calibrated to match the length and volume at the inventory date, but the distribution of ice with elevation is unconstrained. Despite this, the reference volume-elevation distribution of the glacier, based on Huss and Farinotti (2012),

updated to RGI v6.0, is well reproduced at the inventory date. We use two well-studied glaciers to illustrate this (**Fig. 5**), namely the Grosser Aletschgletscher (Valais, Switzerland) and the Mer de Glace (Mont-Blanc massif, France). Also for the other 793 glaciers longer than 1 km, a good match is obtained in general (see section 8).

### 4.3 Glacier dynamics

In our approach the mass transport between grid cells is linearly dependent on the deformation-sliding factor $A$, and is thus important for the ice dynamics. The calibrated values of $A$ for every individual glacier do not have a distinct spatial pattern, nor do they correlate with glacier length or glacier elevation (suppl. mat. Fig. S2). It is not straightforward to compare the values of the deformation-sliding factor to other values from literature used to describe ice deformation and mass transport

(such as rate/creep factors), as different formulations and approaches are utilised, e.g. the inclusion/exclusion of a shape factor, explicit/implicit treatment of basal sliding, different geometry representations, etc. However, it appears that the spread in the modelled deformation-sliding factors, which results from the fact that this value represents several physical processes and uncertainties in our approach, largely falls within the literature range values of rate/creep factors (suppl. mat. Fig. S2). Furthermore, the calibrated median $(1.1 \times 10^{-16}$ $Pa^{-3}$ $yr^{-1})$ and mean $(1.3 \times 10^{-16}$ $Pa^{-3}$ $yr^{-1})$ values are relatively close to the

widely used rate/creep factor from Cuffey and Paterson (2010) based on several modelling studies $(0.8 \times 10^{-16}$ $Pa^{-3}$ $yr^{-1})$.

In the lower parts, where many glaciers have a distinct tongue, a comparison between modelled and observed surface velocities is possible. This is more complicated for the higher parts of the glaciers, where glaciers may be broad and have various branches, which we do not explicitly account for in our approach. In general, our model is able to reproduce the

observed surface velocities for the lower glacier parts despite its simplicity. Based on a set of surface velocity observations from the literature (see **Fig. 6** and suppl. mat. Table S2), a large range of surface velocities, ranging from 1 m yr$^{-1}$ to > 200 m yr$^{-1}$ is well reproduced ($r^2=0.76$; RMSE = 31.8 m yr$^{-1}$), without a tendency for consistent under- or overestimation. This is illustrated for Grosser Aletschgletscher and the Mer de Glace (**Fig. 5**). Some discrepancies may be likely linked to the simplicity of our model and uncertainties in various boundary conditions, but they may in part also be related to the different

time periods between the observations and the modelled state.





### 4.4 Past glacier evolution

Modelled past glacier length and area changes are compared to observations for the time period between the inventory date (typically 2003) and present-day (2017). Periods before 2003 are not considered, as the effect from the imposed 1990 steady

state may still be pronounced on the initial glacier evolution (1990-2003). Furthermore, before the inventory date, length and area changes from the *Δh*-parameterisation (which we apply for glaciers <1 km) are not available, as here the starting point is the observed geometry at the inventory date. Notice that past glacier volume changes are available (e.g. Fischer et al., 2015b), but that these are not used for validation, as they were utilized for calibrating the SMB model component.

**4.4.1    Glacier length**

The modelled length changes between the inventory date (2003) and 2017 are compared to observations for all 52 Swiss glaciers longer than 3 km that are included in the Swiss glacier monitoring network (GLAMOS) (Glaciological Reports, 1881-2017) (**Fig. 7**). Despite the model's simplicity, the general trends in glacier retreat are relatively well reproduced and

there is no general tendency for over- or underestimation. A few outliers exist (highlighted in **Fig. 7**), of which the underestimations can be attributed to a detachment of the lower and upper parts of the glacier, which cannot be captured in our modelling setup. Overestimated retreat rates (Ferpècle, Montminé and Stein) occur for glaciers where the modelled ice thickness in the frontal region at the inventory date is likely to be lower than the reference state and/or where the ice thickness is underestimated in the reference case. When the three glaciers with underestimations due to a disconnection are

omitted, the correlation between the observed and modelled glacier retreat is highly significant (p-value $<1 \times 10^{-3}$; $r^2 = 0.37$). For large glaciers, the retreat is particularly well reproduced: e.g. for glaciers longer than 8 km the root-mean-square error (RMSE) between the observed and modelled 2003-2017 retreat is only 155 m, corresponding to ca. 30% of the mean observed (490 m) and modelled (540 m) retreat over this time period.

**4.4.2    Glacier area**

In the literature, glacier area changes in the European Alps have been derived from various sources. Depending on the time period considered and the ensemble of glaciers studied, estimated glacier area changes vary broadly from $-1.5\%$ yr$^{-1}$ to $-0.5\%$ yr$^{-1}$. Paul et al. (2004) derived area changes for 938 Swiss glaciers and used this to extrapolate a loss of 675 km$^2$ for all

glaciers in the European Alps over 1973-1998/9 period (corresponding to a 22% mass loss, or about $-0.85\%$ yr$^{-1}$ / 26 km$^2$ yr$^{-1}$). For Austria, area changes of $-1.2\%$ yr$^{-1}$ were observed for the period 1998-2004/2012 (Fischer et al., 2015a). On longer time scales, Fischer et al. (2014) derived a relative area loss of 0.75% yr$^{-1}$ for the period 1973-2010 over Switzerland, while for the period 2003-2009 an area loss of 1.3% yr$^{-1}$ was obtained for glaciers in the eastern Swiss Alps. French glaciers lost about one quarter of their area between 1967/1971 and 2006/2009, corresponding to a change of $-0.7\%$ yr$^{-1}$ (Gardent et al.,



2014), while Italian glaciers lost about 30% of their area over the 1959/1962-2005/2011 period (i.e. average of –0.6% yr$^{-1}$) (Smiraglia et al., 2015).

Between 2003 and 2017, we model a glacier area loss of 223 km$^2$ (16 km yr$^{-1}$), corresponding to a relative area loss of 11.3%
(vs. mean area over this time period), or 0.8% yr$^{-1}$. These numbers are difficult to directly compare with values from the literature, as different time periods are considered (implying also a different reference area), and as the area losses strongly depend on size of the considered glaciers (e.g. Paul et al., 2004; Fischer et al., 2014), which also varies between studies. However, a qualitative comparison suggests that the modelled area changes are in general slightly lower than the observations. This discrepancy is mostly related to the fact that many present-day glaciers have frontal regions and ablation
areas with almost stagnant ice, and in some cases also consist of disconnected ice patches, which our model is not able to capture with a simple cross-section parameterisation. By modifying the cross section through increasing λ, a higher modelled area loss is obtained, in closer agreement with observations. However, a higher value of λ may be unrealistic (i.e. produce an area change close to observations for the wrong reasons), and the effect of a different λ value is found to have a very limited effect on the future modelled volume and area changes (this is addressed in section 6.3).

## 5 Future glacier evolution

Our projections suggest that from 2017 to 2050 a total volume loss of about 50% and area loss of about 45% will occur, and that this evolution is independent from the followed RCP (**Fig. 8** and Table 1). This evolution is related to the fact that the
annual and summer temperature differences between the RCPs increase with time and are thus relatively limited in the coming decades (see **Fig. 2**a,b). Furthermore, a part of the future evolution is committed, i.e. being a reaction to the present-day glacier geometry, which is too large for the present-day climatic conditions for most glaciers in the European Alps (e.g. Zekollari and Huybrechts, 2015; Gabbud et al., 2016; Marzeion et al., 2018; cf. discussion in 6.1.1).

By the end of the century the modelled glacier volume and area are largely determined by the RCP that was used to force the climate model (**Fig. 8**). Under RCP2.6, in 2100 about 65% of the present-day (2017) volume and area are lost (–63.2±11.1% and –62.1±8.4% respectively, multi-chain mean, Table 1, **Fig. 8**a). Most of the ice loss occurs in the next three decades, corresponding to about 70-75% of the total changes for the period 2017-2100 (Table 1), after which the ice loss clearly reduces (**Fig. 8**). For an intermediate warming scenario (RCP4.5), by the end of the century about three-quarters of the
present-day volume (–78.8±8.8%) and area (–74.9±8.3%) are lost (**Fig. 8**, Table 1). In contrast to the glacier evolution under RCP2.6, under RCP4.5 a substantial part of the loss takes place in the second part of the 21$^{st}$ century. However, the largest changes still occur in the coming three decades with about 60% of the total changes for the period 2017-2100 (see Table 1). For RCP8.5, the rates of volume loss (–1.5 km$^3$ yr$^{-1}$) and area loss (–25 km$^2$ yr$^{-1}$) are relatively constant until 2070 (**Fig. 8**),





after which they decrease to ca. –0.5 km³ yr⁻¹ and –15 km² yr⁻¹, respectively. By 2100, the Alps are largely ice-free under
RCP8.5, with volume losses of –94.4±4.4% and area losses of –91.1±5.4% with respect to 2017 (see Table 1).

An analysis in which the relative volume loss is compared to present-day glacier characteristics (volume, area, length,
median elevation, mean elevation, minimum elevation, maximum elevation, centre of mass and elevation range) reveals that
under RCP2.6, the elevation range has the highest correlation with the maximum glacier elevation (**Fig. 9** and suppl. mat.
Table S3; $r^2 = 0.57$). The maximum glacier elevation, which is strongly related to the glacier elevation range, also describes
the volume changes well (suppl. mat. Table S3, $r^2 = 0.38$). This also appears from the spatial distribution of the relative
volume loss, where the losses are the most limited for mountain ranges that reach above 3600-3700 m (from West to East):
the Écrin massif, the Mont Blanc Massif, the Monte Rosa Massif, the Bernese Alps, the Bernina Range, in the Dolomites, in
the Ötzal Alps and the High Tauern (**Fig. 9**a). The ice loss is particularly strong under 3200 m a.s.l., where (for a given
elevation band) more than half of the present-day volume disappears by 2100 under RCP2.6 (suppl. mat. Fig. S3). The
remaining ice at these lower elevations is typically from medium-sized and large glaciers, which maintain a relatively large
accumulation area that supplies mass to the lower glacier regions. This is for instance the case for the Mer de Glace (France)
and Grosser Aletschgletscher (Switzerland), where ice is still present below 2500 m a.s.l. by the end of the century under
most EURO-CORDEX RCP2.6 chains (**Fig. 10**a,b). However, both glaciers lose a considerable part of their length
throughout the century, but whereas Grosser Aletschgletscher (**Fig. 10**a) will likely still be retreating by the end of the
century, Mer de Glace will be relatively stable in 2100 under most EURO-CORDEX chains, and may under certain chains
even experience re-advance episodes after 2080 (**Fig. 10**b). Glaciers that spread over a higher elevation range are likely to
suffer even less changes, and in some cases only lose their low-lying tongues (**Fig. 9**b). In contrast, glaciers at low elevation
mostly disappear by the end of the century, even under the moderate RCP2.6 scenario. This is for illustrated for Langtaler
Ferner (Austria), which is situated below 3300 m a.s.l. and is projected to (almost) entirely disappear somewhere between
2050 and 2100 depending on the followed chain (**Fig. 10**c).

The glacier elevation range is also the variable with the highest correlation with respect to the future relative volume changes
under RCP4.5 ($r^2 = 0.63$) and RCP8.5 ($r^2 = 0.51$) (suppl. mat. Table S3). Under these RCPs, except for the related maximum
elevation, also the present-day glacier length correlates significantly with the 2017-2100 volume loss ($r^2 = 0.23$ and $r^2 = 0.20$
respectively). This indicates that under more extreme scenarios the ice loss is very pronounced at all elevations (see also **Fig.
10**d,e) and the remaining ice in 2100 is mainly a relict of the present-day ice distribution, i.e. ice at the end of the century is
only remaining where there is much ice at present at relatively high elevation.





## 6    Discussion

### 6.1 Drivers of future evolution

#### 6.1.1    Committed loss

Part of the future mass loss is committed and related to the present-day glacier distribution of ice. Many glaciers have a present-day mass excess at low elevation, where locally the flux divergence cannot compensate for the very negative SMB, resulting in a negative thickness change (see equation **(5)**) (e.g. Johannesson et al., 1989; Adhikari et al., 2011; Zekollari and

Huybrechts, 2015; Marzeion et al., 2018). To assess the importance of this committed effect, we investigate the glacier evolution under present-day climatic conditions. For this, the model is constantly forced with the mean 1988-2017 SMB (**Fig. 8**). Under these conditions, the glaciers lose about 35% of their present-day volume and area by the end of the century. Simulations with other recent reference periods (e.g. 2008-2017) resulted in relatively similar committed ice losses. This suggests that under RCP2.6, about 60% of the ice losses for the period 2017-2100 are committed losses, while the remaining

40% are related to additional warming.

The committed losses are in agreement with simulations performed by Maussion et al. (2018). In steady-state experiments with the Open Global Glacier Model in which all glaciers, starting from their geometry at the inventory date (typically 2003), are subjected to the 1985-2015 randomised climate, Maussion et al. (2018) project ice volume losses of around 55%

over a 100-year time period for the European Alps. In our simulations, over the period 2000-2100, about 50% of the ice mass is lost for an experiment in which the model is forced with the E-OBS product until 2017 and subsequently with a constant 1988-2017 mean SMB (grey line on **Fig. 8**a).

#### 6.1.2    Role of ice dynamics

Our model setup allows us to analyse the effect of including ice dynamics, compared to the classic GloGEM setup (Huss and Hock, 2015), in which glacier changes are imposed based on the Δh-parameterisation (Huss et al., 2010b) at the regional scale. Comparisons are performed for the period 2003-2100, as the simulations with the Δh-parameterisation start from the geometry at the inventory date (2003 for >96% of all glaciers).

All dynamically modelled glaciers (GloGEMflow) are also run with the Δh-parameterisation (GloGEM). A comparison between the (i) difference in the 2003-2100 relative volume loss (between GloGEMflow and GloGEM) and (ii) various glacier characteristics, reveals that the effect of including ice dynamics is particularly linked to the glacier elevation range ($r^2$ = 0.27; $p<1\times10^{-3}$), and to a lesser extent to other (related) glacier characteristics, such as glacier length ($r^2$ = 0.08), mean





slope ($r^2 = 0.04$), minimum elevation ($r^2 = 0.07$) and the maximum elevation ($r^2 = 0.20$) (all values based on multi-chain mean for RCP2.6). Under RCP2.6, glaciers with a large elevation range (typically >1000 m) experience less loss in the dynamic model on average compared to when being forced with the Δh-parameterisation (**Fig. 11**). The mechanism behind this is the following:

5    (*i*) At the inventory date, the glacier geometry is very similar in both approaches, as the dynamically modelled glacier is as close as possible to the observed geometry (see section 0), which is the starting point for the Δh-parameterisation.

(*ii*) Initially, the total glacier volume evolution is relatively similar in both approaches, as the glaciers are subject to the same climatic conditions and their geometry does barely differ.

(*iii*) However, for glaciers with a large elevation range, relatively more ice is removed at middle and high elevation in the 10   Δh-parameterisation, while in the dynamic model the ice at the lowest glacier elevations is more pronounced.

(*iv*) As a consequence, the geometry starts evolving differently between both approaches, and the larger ice mass at lower elevation in the Δh-parameterisation (and lower ice mass at high elevation) translates into a more negative specific glacier mass balance for the Δh-parameterisation (vs. the dynamic model), resulting in a higher mass loss.

(*v*) In the second half of the 21$^{st}$ century, most glaciers stabilize under a limited to moderate warming (their glacier-wide 15   mass balance evolves towards zero). Given the lower mass and area at middle to high elevations (i.e. around the ELA and higher) for the glaciers modelled with theΔh-parameterisation, these will be slightly smaller to ensure a near-zero SMB.

As glaciers with a large elevation-range are typically the largest glaciers, which make up for a substantial fraction of the total volume, the overall mass loss is thus attenuated when ice dynamics are considered compared to simulations with the Δh-parameterisation (**Fig. 12**, suppl. mat. Table S4). The same holds under RCP4.5 (suppl. mat. Fig. S4a,b), though being less 20   pronounced due to the more intense melting, which also causes glacier changes to occur at higher elevation, and largely disappears under RCP8.5, where the future evolution is largely the same when glaciers are modelled dynamically or with the Δh-parameterisation (suppl. mat. Fig. S4c,d and **Fig. 12**).

### 6.1.3    Role of glacier-specific geodetic mass balance estimation

In this study we use direct geodetic mass balance observations from individual glaciers to calibrate the SMB model component. This contrasts with the original GloGEM setup (Huss and Hock, 2015), in which the calibration is based on regional mass balance estimates. To assess the effect of the SMB calibration source, we perform additional simulations in which the model is forced with a region-wide average geodetic mass balance estimate. In order to allow for a direct 30   comparability, we use a region-wide estimate based on the same geodetic mass balance data as used for our glacier-specific calibration. A value of –0.54 m w.e. yr$^{-1}$ is obtained for the period 1981-2010.

Compared to the reference simulations (with the SMB model calibrated using glacier-specific geodetic mass balances), the simulations in which a region-wide SMB estimate is used for model calibration result in a lower future mass loss (**Fig. 12**,




suppl. mat. Table S4). The difference is the largest under RCP2.6, where the glacier volume change for the 2003-2100 period is –70% (vs. –65% in the standard simulations). The lower mass loss results likely from the fact that for larger glaciers the region-wide SMB estimate is typically higher than their mass balance. By utilising region-wide estimates, the mass balance is thus overestimated in general for these glaciers that make up for a large fraction of the total volume, resulting in a lower

mass loss.

### 6.1.4    Simulated future climate

To assess the role of climate on the modelled future glacier state, we performed a multilinear regression analysis for

categorical data between the RCM chain characteristics (RCP, RCM, GCM and realization) and the glacier volume in 2100. In such an analysis, all independent variables are replaced by dummy indicator variables, which have a value of one when the variable is not considered, and are equal to zero otherwise (e.g. Liang et al., 1992; Tutz, 2012). An analysis in which all possible linear combinations are considered, explains most of the variations in the 2100 volume, as the degrees of freedom are relatively low (cf. suppl. mat. Table S5). An analysis of variance suggests that most of the variance is described by the

RCP (suppl. mat. Table S5; p-value of F-test $<10^{-3}$), as expected, and described earlier (see **Fig. 8**). The only other term that is significant at the 1% level is the GCM ($p<10^{-3}$), followed by the RCM, which is significant at the 5% level (p = 0.04), and finally the realization (p = 0.13) (suppl. mat. Table S5). This indicates that modelled future glacier evolution depends more importantly on the driving GCM than the RCM that is coupled to it, and that the realization has a non-significant effect.

**6.2 Comparison to projections from other Alpine glacier modelling studies**

The future evolution of glaciers in the European Alps has been modelled with models of various complexity and by relying on diverse climate projections. By using a statistical calibrated model in which the ELA is related to summer temperature and winter precipitation, Zemp et al. (2006) estimated an area loss of about 40, 80, and 90% for a respective temperature

increase of 1°C, 3°C and 5°C (2100 v. 1971-1990 mean). Based on 50 glaciers modelled with a retreat parameterisation and subsequent extrapolation Huss (2012) found that between 4% (RCP8.5) and 18% (RCP2.6) of the glacier area will remain by 2100 (vs. 2003). Results from global studies relying on volume/length-area scaling (Marzeion et al., 2012; Radić et al., 2014) and methods in which geometry changes are parameterised (Huss and Hock, 2015) suggest that Alpine glaciers will be subject to volume changes of about –65% to –80% under RCP2.6; between –80% and –90% under RCP4.5; and around –

90% to –98% under RCP8.5 (all values between refer to time period between about 2000 and 2100).

Our simulated volume changes are situated between the lowest projected volume losses (Marzeion et al., 2012) and the highest projected volume losses (Huss and Hock, 2015), and are relatively close to the estimates of (Radić et al., 2014) (**Fig. 13**; all changes are considered over the same reference period as study from the literature). Given the different models and





inputs, a direct comparison is however difficult with the results of Marzeion et al. (2012) and Radić et al. (2014). Differences in initial volume estimates may play an important role (e.g. Huss, 2012), and so does the climatic forcing and translation into mass balance, which is study-dependent. The lower losses compared to the results of GloGEM (Huss and Hock, 2015) suggest that the effect of including ice dynamics (reducing the mass loss, section 6.1.2), combined with a slightly lower

temperature increase (from EURO-CORDEX RCM ensemble vs. CMIP5 simulations over Europe used in Huss and Hock (2015)), dominate over the effect of using glacier-specific geodetic mass balances (section 6.1.3)

To our knowledge, three studies have been performed in which the future evolution of an individual Alpine glacier is simulated with 3-D simulations accounting for longitudinal stresses, i.e. with higher-order and full-stokes models (Jouvet et

al., 2009, 2011; Zekollari et al., 2014). Simulations with our flowline model agree well with those for the Rhonegletscher and Grosser Aletschgletscher (Jouvet et al., 2009, 2011), and project a slightly higher mass loss compared to those for the Vadret da Morteratsch complex (Zekollari et al., 2014). More details are provided in the supplementary material (Table S6). Given the differences in boundary conditions and model specifications (e.g. bedrock geometry, SMB model etc.) these findings should not be overinterpreted, but give a qualitative indication that our model is able to relatively well reproduce

changes obtained from more complex and detailed studies on individual ice bodies.

### 6.3 Sensitivity experiments and uncertainty analysis

#### 6.3.1    1990 steady state assumption and deformation-sliding factor

In this study, we opted for a 1990 steady-state glacier, as the glaciers in the European Alps were generally not too far off equilibrium around this period, with SMB conditions for many glaciers being close to zero (Huss et al., 2010a; WGMS, 2018). By imposing a steady state in 1990 (through an offset in the 1961-1990 climatic conditions), the glacier length at the inventory date can be influenced. By relying on an earlier time (before 1990) for the steady state, in some cases the steady-

state glacier geometry does not determine the glacier length at the inventory date anymore, as the period between the steady state and the inventory date exceeds the typical Alpine glacier response time of several years to a few decades (e.g. Oerlemans, 2007; Zekollari and Huybrechts, 2015). In such a case, only the glacier volume at inventory date can be matched, through a modification of the deformation-sliding factor.

In order to assess the effect of the 1990 steady state assumption and the specific calibration procedure utilized in this study, we performed alternative simulations starting in 1950, in which only the volume at the inventory date is matched (no check on glacier length). Through this approach, the calibrated deformation-sliding factor is lower than in the two-step approach used as the reference (mean value of $0.6 \times 10^{-16}$ Pa$^{-3}$ yr$^{-1}$ vs. $1.3 \times 10^{-16}$ Pa$^{-3}$ yr$^{-1}$; see suppl. mat. Fig. S2), and as such, this experiment also provides an insight into the effect of variations in the deformation-sliding factor on future evolution. This is



furthermore of interest, as the deformation-sliding factor depends on the reference glacier volume, which is itself a model result (Huss and Farinotti, 2012) with its own uncertainties. The lower deformation-sliding factors (vs. the two-step calibration approach) result in slightly shorter glaciers at present (vs. observations), as they represent the same volume at the inventory date. As a consequence, the glaciers are located slightly higher and have a somewhat less pronounced future ice

loss (**Fig. 14**). Despite this, the effect on future evolution is rather limited: under RCP2.6 the 2017-2100 the difference in computed volume change is in the order of 5% between classic volume-length calibration and the 'volume only calibration'. Under RCP4.5 and RCP8.5 the differences in calibration procedure and rate factors barely translate in a different 2100 volumes.

**6.3.2    Glacier cross section**

In all simulations, a trapezoidal cross section with an angle $\lambda$ of 45° was used (suppl. mat. Fig. S1). Simulations with a very pronounced trapezium shape ($\lambda$= 80°), cf. a V-shaped valley, result larger area changes for the period 2003-2017 of –1.2% $yr^{-1}$, which is in better agreement with observations (–0.8% $yr^{-1}$ in classic case) (**Fig. 14**b). However, on the longer term, the

effect on the volume and area loss is very limited, and the area in 2100 is only slightly lower compared to the standard run ($\lambda$= 45°), typically in the order of 2-3% (vs. present-day area, **Fig. 14**b). The same holds for the volume, which is about 2% lower (vs. present-day volume **Fig. 14**a). In the case a rectangular cross section is used ($\lambda = 0°$), the differences in projected volume and area changes are also very small (in the order of 1-2%) compared to standard run ($\lambda$= 45°). This is in line with the results obtained in the original GloGEM study (Huss and Hock, 2015) on the global scale, where sensitivity tests with

other cross-sectional shapes suggested that projected mass losses would may decrease/increase by 1-4%.

In general, our results indicate that the differences in projected volume and area changes from the various RCM chains (for a given RCP) are much larger than differences obtained from model parameters. This is in agreement with other glacier evolution studies, as for instance highlighted by Goosse et al. (2018) on the centennial glacier length fluctuation modelling

of an ensemble of alpine glaciers with OGGM, and by Marzeion et al. (2012), who also find that the ensemble spread within each RCP is the biggest source of uncertainty for the modelled future mass changes.

**7    Conclusions and outlook**

In this study, we extended an existing glacier evolution model (GloGEM) through the incorporation of an ice flow component. The so-extended model, GloGEMflow, was used to simulate the future evolution of individual glaciers in the European Alps. In contrast to previous simulations over the European Alps, we used a glacier-specific geodetic mass balance estimate for model calibration. A new initialisation procedure was proposed, in which model parameters were calibrated to match the reference glacier length and volume at the inventory date. This novel model setup and its calibration were



validated with a broad range of in-situ data, including SMB measurements, glacier length changes, glacier area changes and ice surface velocity measurements.

The calibrated model was used to simulate the future evolution of the glaciers in the European Alps under high-resolution

RCM future climate scenarios from the EURO-CORDEX ensemble. These simulations of future glacier change can be of interest for various applications (e.g. runoff projections, hydroelectricity production, natural hazards, touristic value, etc.) and are available in an online repository for every individual glacier (see section 8 for details). Our simulations indicated that under RCP2.6, by 2100 about a two-thirds of the present-day glacier volume (–63±11%) and area (–62±8%) will be lost. Under a strong warming, the European Alps will be largely ice-free by the end of the century, with projected volume losses

of –79±9% under RCP4.5 and –94±4% under RCP8.5 (2017-2100 period). The future glacier evolution is mostly controlled by the imposed RCP. For a given RCP, the spread in future projections from different model chains is mainly determined by the driving GCM (rather than the RCM coupled to it), and was found to be much larger than the differences resulting from model parameter variability.

This study focused on the European Alps, for which a vast dataset on glacier data is available. By relying on this unique dataset and by combining it with a novel glacier modelling setup, we were able to quantify a part of the uncertainties related to assumptions that are widely used in regional and global glacier modelling studies, such as the use of region-wide SMB estimates for model calibration and the implicit treatment of ice dynamics. The inclusion of ice dynamics reduced the projected ice loss compared to simulations relying on a retreat parameterisation and this effect was found to be particularly

important for glaciers that extend over a large elevation range. This implies that the inclusion of ice dynamics is likely to be important for global glacier evolution projections, indicating that there is still a relatively large potential to improve these projections.

## 8   Data availability

The following material will be available in an online repository once/if this study is accepted for publication: (1) modelled glacier geometries at inventory date for all dynamically modelled glaciers as individual figures, and (2) modelled future (2017-2100) glacier volume and area evolution for every individual glacier (multi-model mean for RCP2.6, RCP4.5 and RCP8.5) as comma-separated value files (.csv). All other data presented in this paper will be available upon request.

**Acknowledgments**

H. Zekollari acknowledges the funding received from WSL (internal innovative project) and the BAFU Hydro-CH2018 project. We thank the data providers in the ECA&D project (http://www.ecad.eu), which contributed to the E-OBS dataset.



We are also grateful to the groups who participated in the coordinated regional climate downscaling over Europe (EURO-CORDEX) for making their data available. Our thanks also go out to the World Glacier Monitoring Service (WGMS) for coordinating and distributing SMB data from various groups and to the Swiss Glacier Monitoring Network (GLAMOS) for making glacier length changes available. A. Bauder is thanked for helping retrieving surface velocity measurements.

## Appendix A: Model initialisation

As a first guess a deformation-sliding factor of $1 \times 10^{-16}$ Pa$^{-3}$ yr$^{-1}$ is used ($A_1$). This is combined with a SMB bias, which is expressed as a change in ELA ($\Delta ELA_1$) and chosen in order to ensure a zero mass balance over the present-day glacier

geometry. These parameter values are imposed until a steady-state glacier is obtained, and this geometry is subsequently used to model the glacier evolution between 1990 and the inventory date (typically 2003) (**Fig. 3**). After the first step, a glacier with a volume $V_1$ is obtained. Subsequently, this setup is repeated by modifying the deformation-sliding factor, until the reference glacier volume at the inventory date ($V_{ref}$) is matched (within 1%). The second guess for the deformation-sliding factor ($A_2$, i.e. second step of optimization procedure) corresponds to:

$$A_2 = A_1 \left( \frac{V_{ref}}{V_1} \right)^{-4} \tag{8}$$

Subsequent guesses of $A$ are derived from a polynomial fit between glacier volume (independent variable) and all previous estimates of $A$ (dependent variable). The order of this polynomial fit corresponds to the number of previous guesses – 1, e.g. the third guess for $A$ relies on the previous two attempts, for which a first order polynomial (i.e. a linear function) is constructed. This leads to a quick convergence to reference glacier volume at inventory date, typically within 3-4 attempts.

Once a match for the glacier volume is obtained, a check on the glacier length is performed. If the glacier length ($L_1$) deviates more than 1% from the reference glacier length ($L_{ref}$), the volume calibration is reapplied, for which the ELA bias ($\Delta ELA_1$) is increased/decreased with 10 m. The volume calibration is performed again (see above), where the first guess for the deformation-sliding factor is now equal to the last guess that resulted in a volume match. Once the volume is matched (typically within one or two attempts), a new check on the glacier length at inventory date is performed. If the length is not

matched at the second attempt, from the third attempt onwards the $\Delta ELA$ is estimated based on a linear fit between the previous two attempts (independent variables) and the glacier length (dependent variables), or a shift of 10 m if both attempts resulted in the same glacier length (which can occur due to the discretization of the glacier geometry). All together, this methodology results in a fast convergence, and in general the entire simulation (creating steady state and running glacier from 1990 to inventory date) needs to be performed about 3-10 times. This takes on average about 10-20 seconds per glacier

on a single core on a modern laptop.



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





**Tables**

| | Volume in km$^3$ (and relative change vs. 2017) | | Area in km$^2$ (and relative change vs. 2017) | |
|---|---|---|---|---|
| | 2050 | 2100 | 2050 | 2100 |
| **Committed loss** | 71.4 ( -25.9 ± 7.4%) | 60.8 ( -36.9 ± 6.3%) | 1277.9 ( -23.3 ± 7.7%) | 1091.2 ( -34.5 ± 6.5%) |
| **RCP 2.6** | 51.7 (-47.0 ± 10.3%) | 35.9 (-63.2 ± 11.1%) | 1037.6 (-43.9 ± 9.7%) | 701.7 (-62.1 ± 8.4%) |
| **RCP 4.5** | 50.0 (-48.8 ± 9.2%) | 20.7 (-78.8 ± 8.8%) | 1006.9 (-45.6 ± 8.0%) | 464.2 (-74.9 ± 8.3%) |
| **RCP 8.5** | 47.1 (-51.8 ± 11.5%) | 5.4 (-94.4 ± 4.4%) | 948.2 (-48.8 ± 9.2%) | 165.4 (-91.1 ± 5.4%) |

**Table 1:** Overview of multi-chain mean future glacier evolution based on RCM chains from the EURO-CORDEX ensemble.

The committed loss corresponds to the loss obtained by permanently applying the mean SMB obtained from 1988-2017

5    climatic conditions.



**Figures**

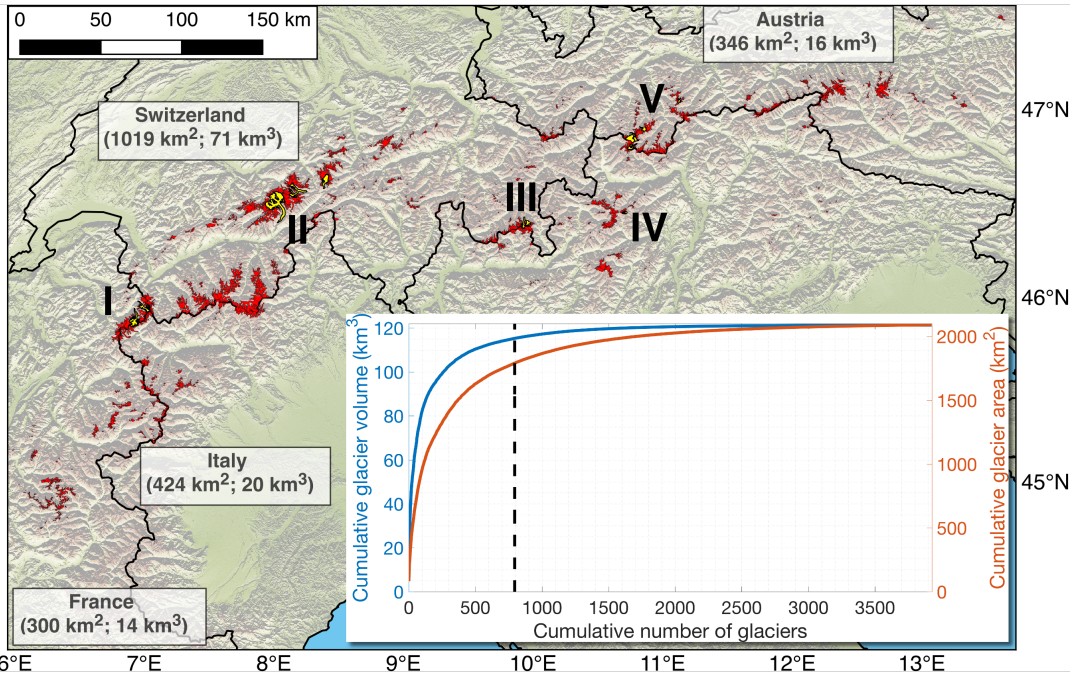

**Fig. 1.** Distribution of glaciers (red areas) in the European Alps. Outlines correspond to the glacier geometries at the
5   Randolph Glacier Inventory (RGI v6.0) date (typically 2003) (Paul et al., 2011; RGI Consortium, 2017). The hill shade in
the background is from the Shuttle Radar Topography (SRTM) DEM (Jarvis et al., 2008). Glaciers discussed in this
manuscript and the supplementary material are highlighted in yellow: (I) Mer de Glace & Argentière; (II) Grosser Aletsch,
Unteraar & Rhone, (III) Morteratsch, (IV) Careser, (V) Hintereisferner, Kesselwandferner, Taschachferner, Gepatschferner
and Langtaler Ferner. The inset shows the cumulative glacier area and volume, sorted by decreasing glacier length. The
10   dotted line is the division between glaciers longer (left) and shorter (right) than 1 km. Glacier area is from the Randolph
Glacier Inventory (RGI v6.0) (RGI Consortium, 2017); volume and length are as derived by an updated version Huss and
Farinotti (2012).

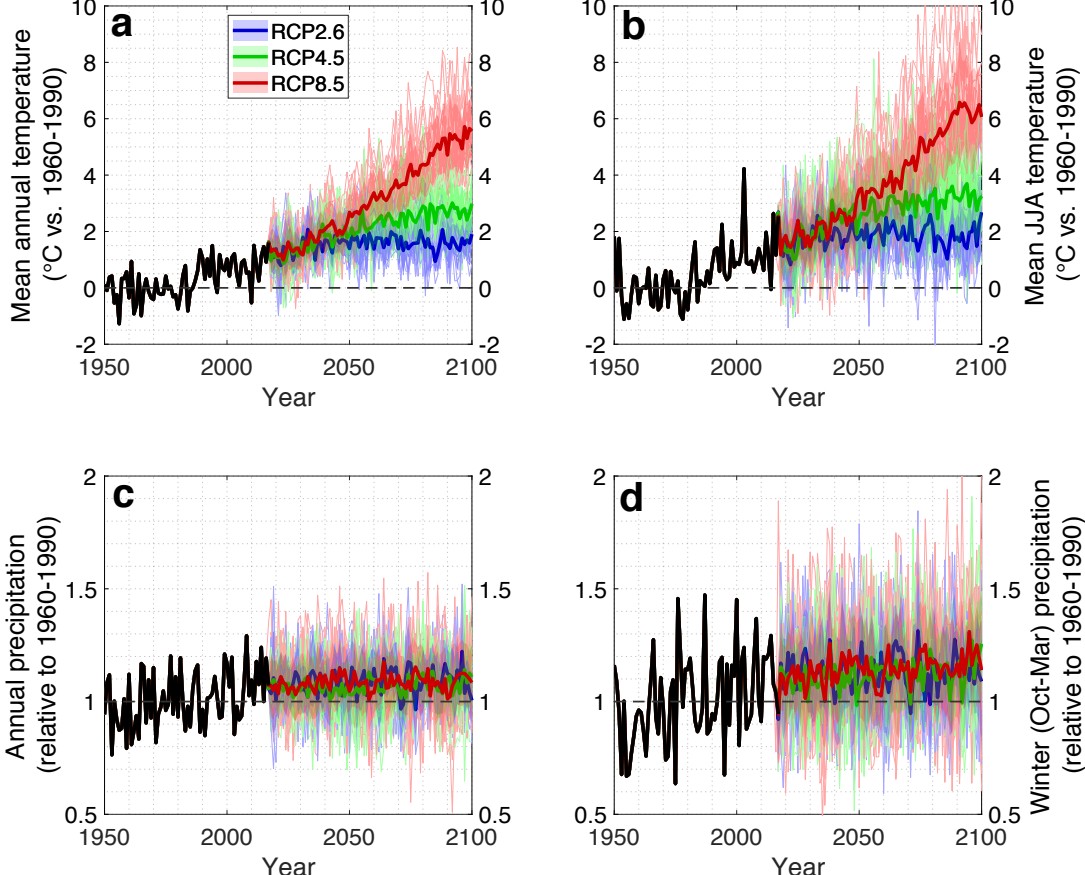

**Fig. 2.** Debiased temperature anomaly (a: annual & b: June-July-August) and debiased precipitation anomaly (c: annual & d: October-March) between 1950 and 2100 with respect to 1961-1990 (horizontal dotted line). All values correspond to the mean over all grid cells used in this study. The thick black line represents the evolution of the variables for observational period (E-OBS dataset). The coloured thin lines represent the evolution for individual model chains from the EURO-CORDEX ensemble (51 in total, see suppl. mat. Table S1), the thick lines are the chain means (one per RCP). Transparent bands correspond to one standard deviation.





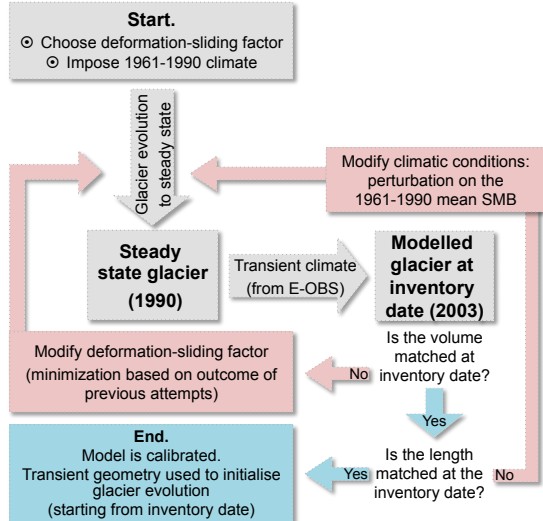

**Fig. 3.** Model initialization for creating a glacier with the reference length and volume at the inventory date.



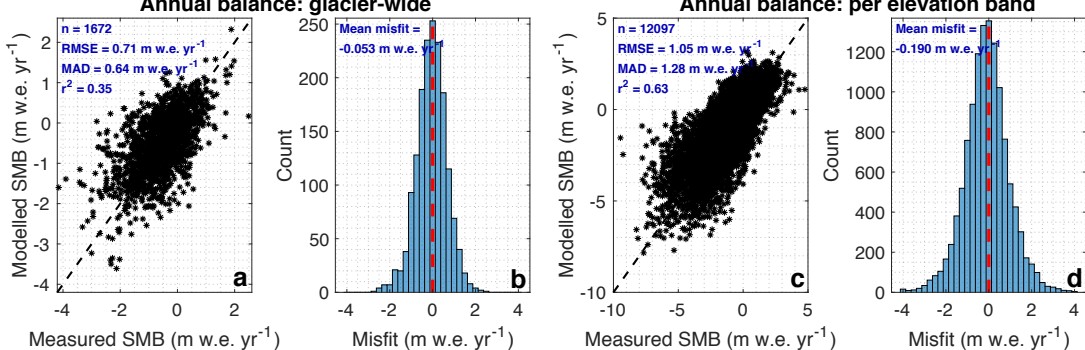

**Fig. 4.** Evaluation of modelled SMB against observations from the WGMS (2018) database. Scatterplots (a & c) and frequency of misfits (b & d) of modelled vs. measured glacier-wide annual balances (a & b) and annual mass balances per elevation band (c & d). Dashed red lines in panels b & d represent the zero misfit. In panel a & c, n corresponds to the number of observations, RMSE is the root-mean-square error, MAD is the Median Absolute Deviation and $r^2$ is the correlation coefficient.





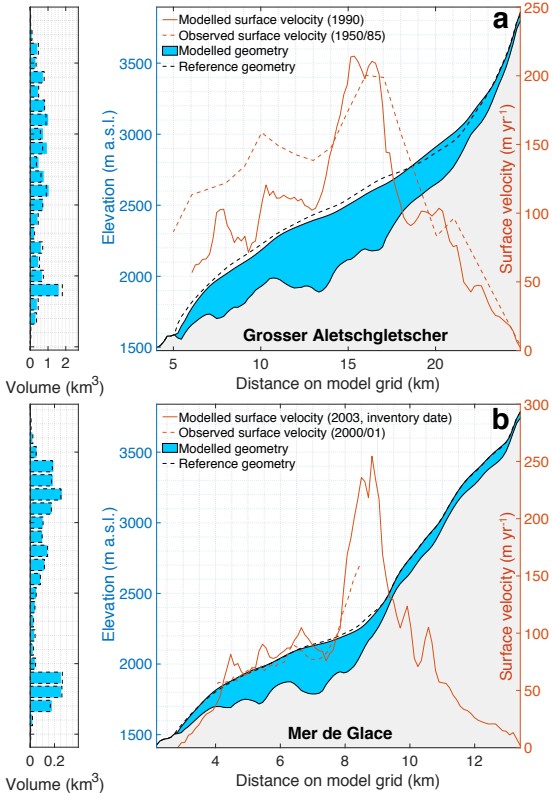

**Fig. 5.** Comparison between reference and modelled (i) glacier geometry, (ii) volume-elevation distribution and (iii) surface velocities for Grosser Aletschgletscher (a) and Mer de Glace (b). Reference geometries and volume elevation distribution are at inventory date (2003) and based on Huss and Farinotti (2012). Observed surface velocities for Grosser Aletschgletscher are from Zoller (2010), and correspond to a 1950/1985 point averages, while observed velocities for the Mer de Glace are derived from 2000/2001 SPOT imagery (Berthier and Vincent, 2012).





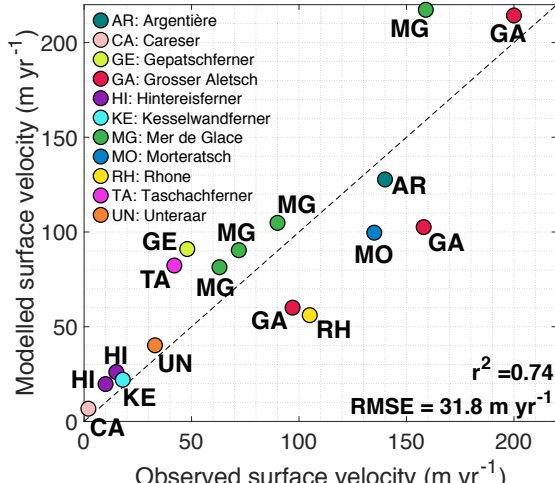

**Fig. 6.** Observed vs. modelled surface velocities for selected glaciers in the European Alps. For some glaciers several data points exist, consisting of different locations on the glacier. More information concerning the surface velocities and corresponding references are given in the supplementary material (Table S2). For glacier location, see **Fig. 1**.





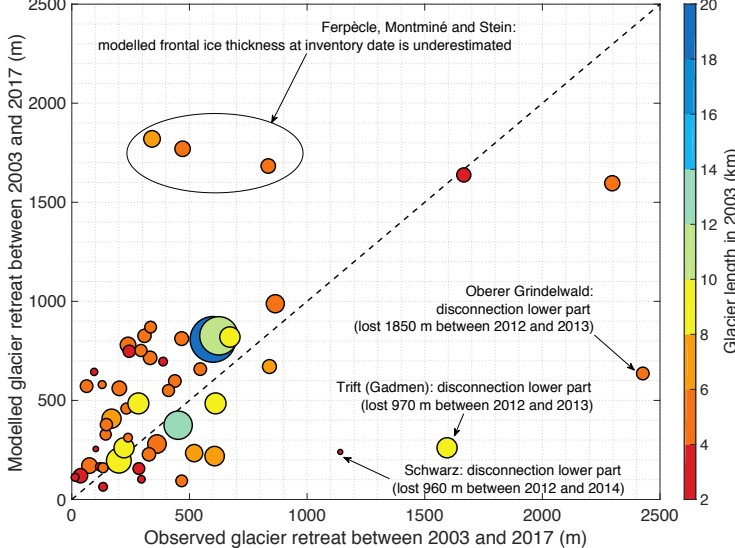

**Fig. 7.** Observed vs. modelled glacier retreat (length change) between 2003 and 2017 for all 52 glaciers longer than 3 km monitored by GLAMOS (Glaciological Reports, 1881-2017). Point size is proportional to glacier area (cf. colour bar).





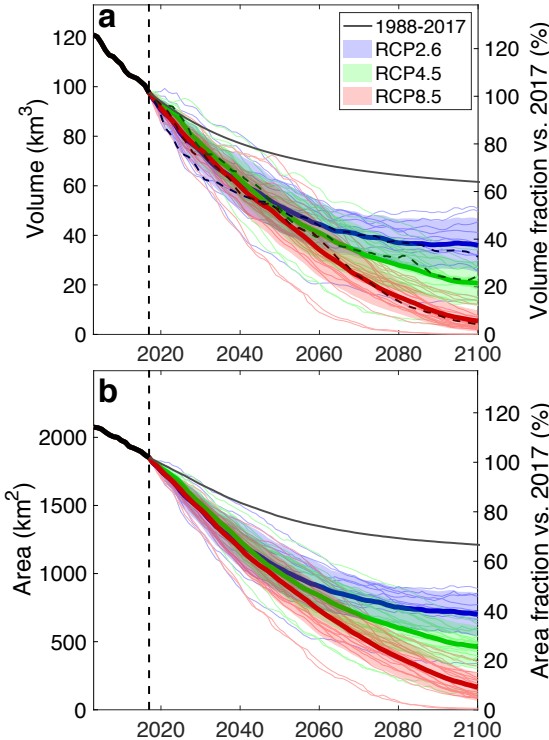

**Fig. 8.** Ensemble (a) volume and (b) area evolution for various EURO-CORDEX RCM simulations and committed loss
(mean 1988-2017 conditions). Thin lines are individual model chains (51 in total, see suppl. mat. Table S1). The thick line is
the RCP mean and the transparent bands correspond to one standard deviation. In panel (a), the coloured dotted lines
correspond to the model chains that are closest to the multi-model mean. The vertical dotted line represents the year 2017
and marks the transition between E-OBS and EURO-CORDEX forcing.





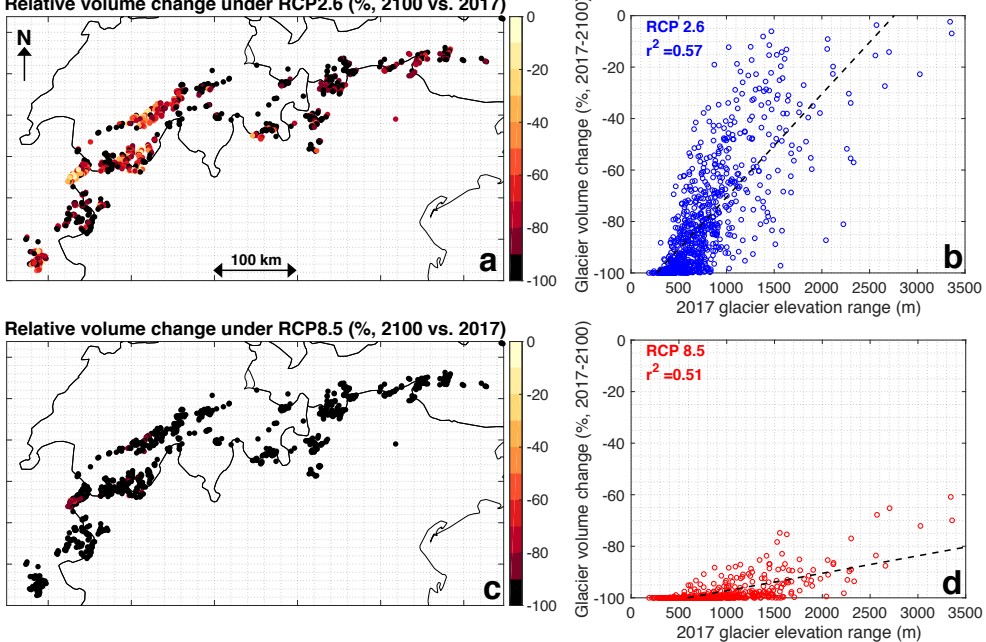

**Fig. 9.** Relative volume changes between 2017 and 2100 under RCP2.6 (multi-chain mean, panel a&b) and a RCP8.5 (multi-chain mean, panel c&d). Results are shown for all 795 glaciers for which the future evolution is simulated with the dynamic model. Panel (b) and (d) represent the volume change as a function of the two present-day glacier elevation range.





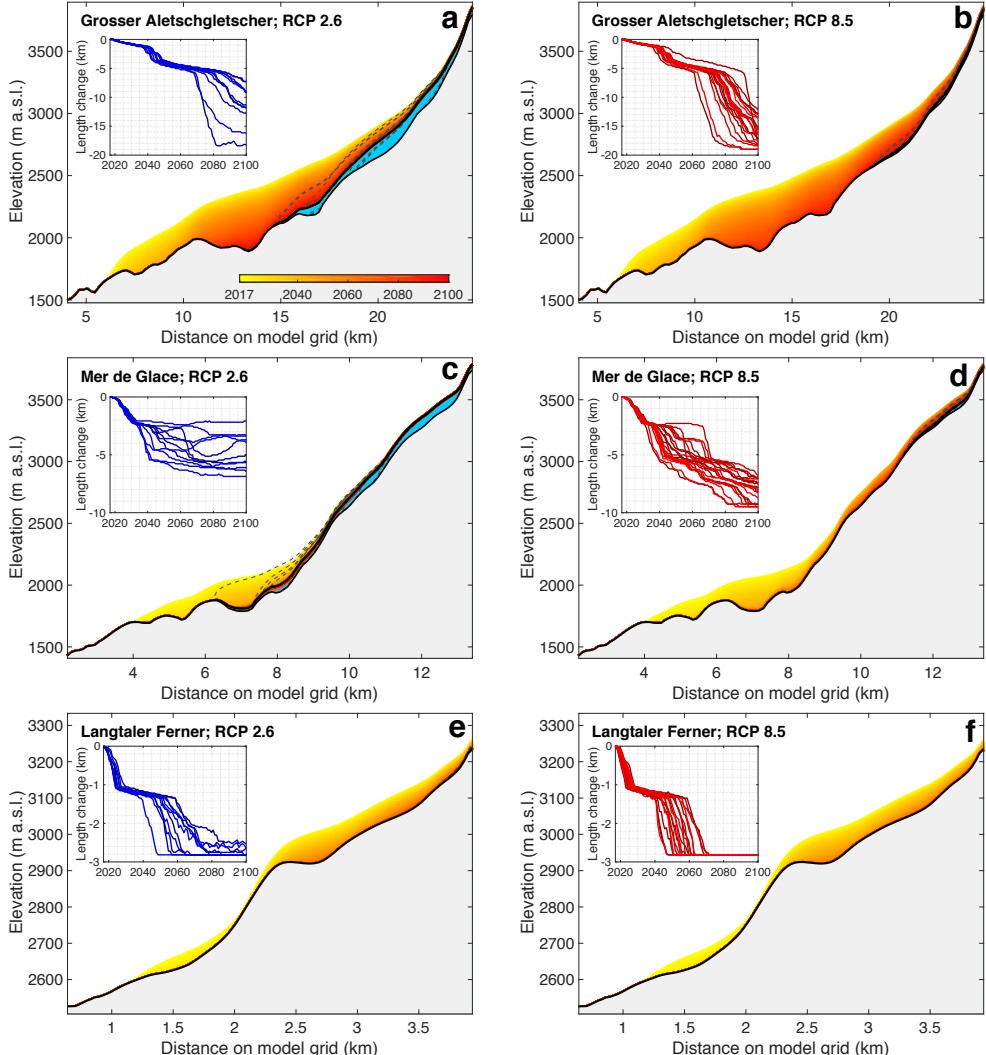

**Fig. 10.** Future evolution of the Grosser Aletschgletscher (a,b), Mer de Glace (c,d), and Langtaler Ferner (e,f) under RCP2.6 (a,c,e) and RCP8.5 (b,d,f). The 2017-2100 evolution corresponds to the multi-model mean surface evolution, while the blue area is the multi-model mean glacier geometry at the end of the century. The dotted lines represent the 2100 glacier geometries for individual model chains (cf. suppl. mat. Table S1). The insets represent length changes over the 2017-2100 time period for every individual RCM chain.





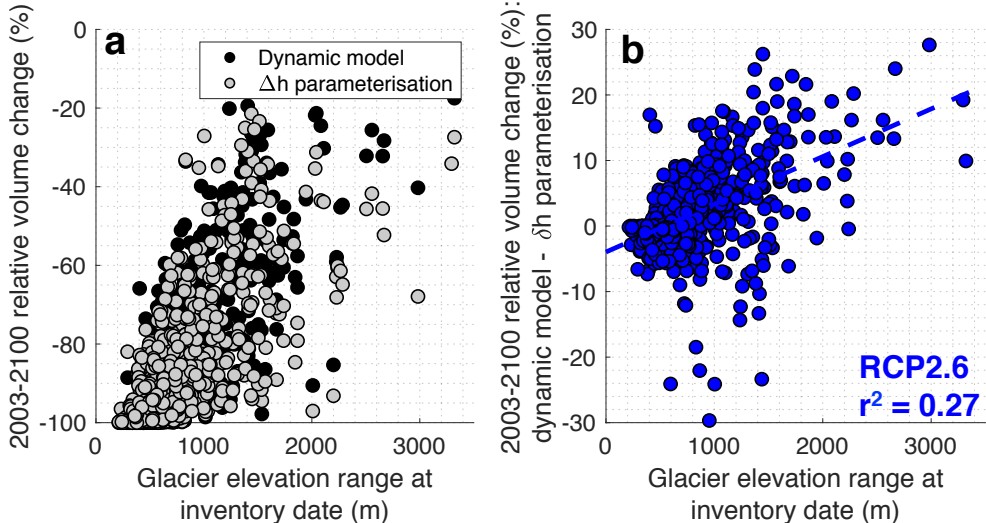

**Fig. 11.** Future glacier evolution under RCP2.6 for individual glaciers with the dynamic model and corresponding glacier simulation with the Δh-parameterisation. All values correspond to RCP2.6 multi-chain mean values.

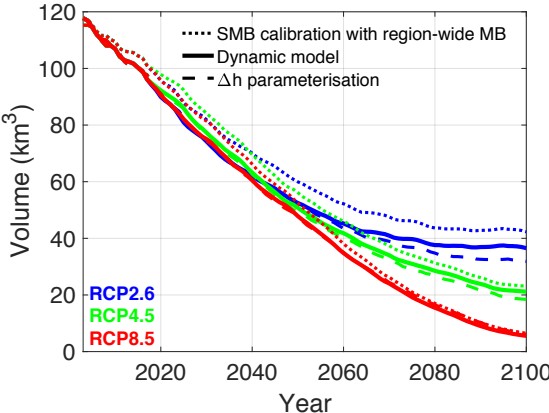

**Fig. 12.** Future glacier volume evolution as simulated with (i) the dynamic model forced with an SMB calibrated to individual glaciers (standard run), (ii) the Δh-parameterisation (Huss et al., 2010b), and (iii) the dynamic model, where the SMB model component is calibrated with a region-wide MB estimate. Results are shown for glaciers longer than 1 km at inventory date and correspond to the multi-chain values from RCM chains from the EURO-CORDEX ensemble (for a given RCP).



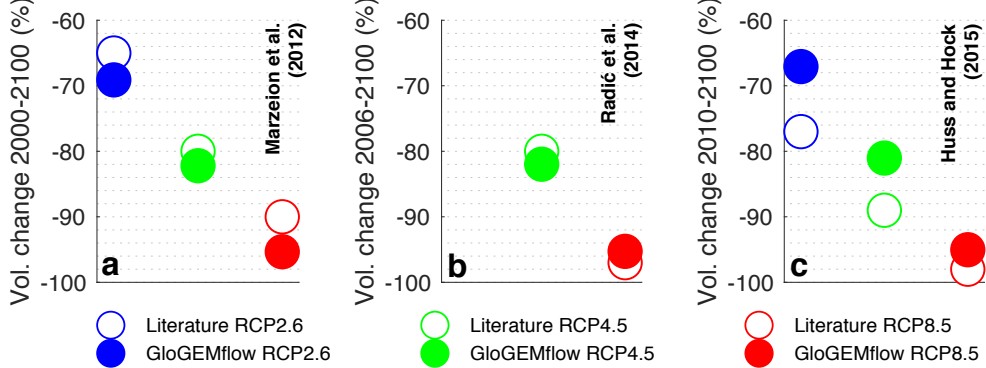

**Fig. 13.** Modelled volume changes and comparison with values from the literature (Marzeion et al., 2012; Radić et al., 2014; Huss and Hock, 2015). The considered time period is in line with the considered study and spans from the early 21$^{st}$ century to 2100.



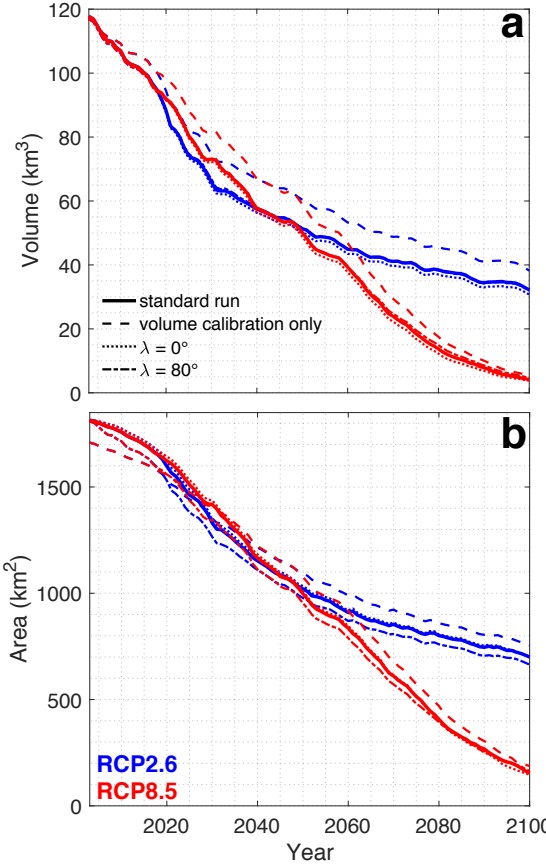

**Fig. 14.** Sensitivity of volume (a) and area (b) for different cross-sectional geometries and under a different calibration procedure. Results are shown for all 795 glaciers for which the future evolution is simulated with the dynamic model. The standard calibration with $\lambda = 45°$ corresponds to the classic setup. The colours correspond to different RCPs. Only the model

5   chain closest to the multi-model mean volume evolution is shown (dotted line in **Fig. 8**a; see also suppl. mat. Table S1).