# Peer review of "Modelling the future evolution of glaciers in the European Alps under the EURO-CORDEX RCM ensemble"

_The Cryosphere, 2018_

## Referee Comment (RC1) · Marzeion (Referee) · 21 Jan 2019

Zekollari and colleagues present projections for glaciers in the European Alps using a glacier model that explicitly resolves ice dynamics and is forced by GCM projections dynamically downscaled in the framework of CORDEX. The manuscript presents a timely step forward for modeling of glaciers on large regional scales. It is written well and succeeds to make the assumptions and limitations of the model accessible. I am particularly impressed with high quality of the presentation of the results in the Figures. Overall, I have no doubt that this is a valuable contribution to the literature and eventually should be published. However, there are a number of minor points that

should be addressed by the authors, as well as three somewhat major issues:

Major comments:

Use of CORDEX data: This is the biggest issue I have: the authors use the projections from CORDEX, present a relatively thourough validation (some comments on this below), but do not at all address a question that seems very obvious to me: from the perspective of modeling glaciers, is there added value in CORDEX? I.e., what difference does it make applying the downscaled data opposed to applying the GCM data directly? I'm aware that because of internal variability, there is not much use in using the CORDEX (or GCM) data during the 2003-2017 validation period in the same way the E-OBS data are used. But there is still some insight to be gained concerning, e.g., temporally accumulated area and volume loss. Also: through comibination with differen RCMs, the ensemble size of CORDEX is increased - however, there are GCMs not part of CORDEX. Taking the ensemble as a measure of uncertainty, how does the addition of the RCM "axis" affect the ensemble uncertainty? Is is equivalent to taking a larger GCM ensemble, or can insights be gained through the downscaling? Of course, this point is not a weakness of the manuscript. But I see the approach taken by the author as great opportunity to address some questions that are very relevant, since dynamical downscaling is very expensive, and its value is somewhat debated. I think it would be great if the authors took this opportunity.

Initialization: Why is glacier length chosen instead of glacier area? It seems to me that area is better constrained than glacier lenght, since the length is to some degree an arbitrary choice resulting from the representation of the flow line. Also, using the 1961-1990 climatology, was variability preserved? Where the forcing timeseries during that period detrended? As the authors say, there is no assumption about an equilibrium state of the glacier, but an assumption that the glacier woud be able to undergo the transition from an equilibrium state in 1990 to the obsserved transient state in (typically) 2003). Doesn't this correspond to an assumption that the response time of the glaciers is shorter than 13 years? You discuss this in Sect. 6.3, but it would be good to have

some arguments presented already in Sect. 3.3.

Validation: The geodetic mass balances used for calibration are most probably not in-dependent of the in situ measurements used for validation. For those glaciers that have geodetic observations that do not temporally overlap with the in situ measurements, it would be good to recalibrate using these, and revalidated using the temporally independent in-situ values. I'm also wondering about the choice of using geodetic observations for calibration, and in-situ for validation. The opposite woud seem like the more natural choice to me, since geodetic MBs can cover longer time periods, and thus allow to include effects from ice dynamics in the total uncertainty. Is there a specific reason not to make the calibration based on in-situ observations, and validate using geodetic MBs?

Minor comments

Main Text:

P1 L10: ... ice flow processes, _of_ which the latter ...

P1 L10: you could specify it is not included explicitly, since the (simple) parameterizations used are supposed to parameterize its effect.

P2 L21: specify "glacier model parameters".

P2 L17: either delete "using various methods", or briefly specify.

P2 L18 (and P15 L27): volume/length/area or volume-lenght-area scaling

P2 L21: I suggest RCP 8.5 should not be called "extreme", since the different scenarios are not based on probabilities, nor is there a physical reason to view RCP8.5 above some limit value.

P3 L2-4: Not sure why glaciers in the Canadian Rockies should not be controlled by topography and local effects. Either specify, or delete; I think there is value in different approaches to similar problems by itself, so I don't see a need for a very

strong justification of your study here.

P3 L9: you might also refer to Goosse et al. (2018), which has transient simulations (DOI: 10.5194/cp-14-1119-2018)

P4 L7: delete "in these little-known connections" (or specify why they are lesser known than the rest of the glacier).

P4 L12: I would prefer "Climate data" or "Atmospheric boundary conditions"- but that is maybe a matter of taste.

P4 L19: "... has a higher resolution than the reanalysis data used in Huss & Hock (2015, ERA-INTERIM) and goes back further in time."

P4 L21: rephrase around "several chains used for", since it is not quite clear here what "chains" you are referring to.

P4 L28 (and throughout the study): perhaps it would be better to call them "model combinations" than "chains" (these "chains" have only two links anyway...).

P4 L30: "... a peak-and-decline scenario ..."

P4 L32: "Note that while country-specific ..."

P4 L32: "... such as the projections recently released for the CH2018 report, ..." (or something similar - i.e., distinguish between scenarios and projections).

P5 L13: because of the difference in resolution between E-OBS and the RCMs, there is probably not a direct equivalent in the grind points. Please specify how you handle this.

P6 L13: "closest to the glacier" - there are probably glaciers that are covered by multiple grid cells - are then multiple grid cells taken into account, or are you using the glacier center point? Please specify.

P6 L15 (and L19): "temperature-index melt model"

P6 L26-30: While precipitation is the least constrained boundary condition, the degree-day factors are probably the most potent paramteters for tuning. Why is the order chosen like this, and how is the defaul degree-day factor chosen? Please specify.

P8 L25: It is fine to seperate the validation of the SMB model from the ice dynamics, but there should not be much difference if the dynamics are behaving well. Also, the SMB observations are probably not taking into account the RGI outlines, such that there is some disagreement anyway. Have you tested how much the results differ if you switch on ice dynamics?

P8 L29-30: I don't agree: this correlations measures the skill of the model to reproduce more positive mass balances at the upper part of the glacier than at the lower parts. A better measure for the SMB model's ability to represent the MB profile would be a comparison of the (temporally averaged) MB profiles, or correlating the deviations from the "climatological" (i.e., mean) profile.

P10 L20: delete "1x".

P10 L27: delete "In the literature".

P10 L30: "over the period 1973-1998/9".

P11 L19: "evolution is independent of the scenario".

P12 L6: I don't understand this sentence, please rephrase. Do you mean volume change?

P12 L11: "strong below 3200 m".

P12 L18: delete "may" (you already say it's only under certain model combinations).

P12 L21: "This is illustrated for Langtaler ..." Sect. 6.1.1: Just out of curiosity, it would be interesting to know about the committed mass loss past 2100.

P14 L6: correct Section number.

Appendix:

P19 L16: delete "1" at the end of the line.

Tables:

Table 1: The "committed loss" line is a bit confusing, since the committed loss should not be time-dependent. Perhaps you can call this, e.g., the "realized fraction of committed loss"?

Figures:

Generally - but particularly Fig. 9: I'm not a big fan of grids in figures. If there is a need to read numbers/differences from the figures, these numbers should be mentioned on the figure or in the text, and no grid would be necessary; if there is no such need, the grid only adds clutter. Please consider removing grids.

P 27 L3: "relative to 1961-1990" (instead of "with respect to").

P27 L4: is that the mean of all "alpine" EUROCORDEX grid cells, or just the ones that contain glaciers? If the latter, I think it would make more sense to also weight them by glacier area.

Fig. 2: I have a hard time seeing the transparent bands; I suggest deleting them, since that information is already shown in the distribution of the individual (thin) lines.

Figs. 5, 6: are there uncertainty estimates available for the observed velocities? If so, it would be good to include them (e.g., just a single error bar so that the observation uncertainty can be compared to the differences with to modeled velocities).

---

## Referee Comment (RC2) · Maussion (Referee) · 28 Jan 2019

**1    General comments**

The manuscript by Zekollari and colleagues presents new estimations of projected glacier change in the European Alps. It is a well conducted study, the paper is well written and the results are interesting. The inclusion of ice dynamics, the use of CORDEX data instead of coarser GCMs and the large amount of calibration data are the main novel points in this study.  It will become the new reference study for future glacier change in the European Alps and as such, it is likely to receive a larger interest from

the general public and the media.

I have two major concerns that need to be addressed before publication, as well a several specific questions / recommendations.

**1.1 Validation and uncertainty**

I acknowledge the efforts realized to use so many different observational datasets, an exercise only possible in the European Alps. However, I have several issues with the model validation in this study:

1. there is no information about how many glaciers (and how much ice area/volume) are simulated without any calibration data. For these glaciers, the geodetic MB is "interpolated" and the effect of this interpolation is not assessed

2. there is no indication as to the computation of the uncertainty ranges provided in the glacier changes (e.g. in the abstract). Does it originate from the forcing ensemble? The model RMSE? It is hard make further assessments without this information

3. the validation using observed traditional MB does not make sense to me, because the model has been calibrated on geodetic MB on the same glacier (both data are not exactly the same, but close - see also the comment of Ben Marzeion). If anything, you should use cross-validation here: when assessing the performance of the model on a given glacier, you remove the selected glacier from the calibration dataset, then use this data plus the traditional MB measurements to assess the model. This would also help to address point 1

4. it is problematic that the effect of the RCM forcing is not assessed at all. The plots all start in 2017, so any sceptic reader could say: "this is all extrapolated without

test in the past". I understand the problems behind the validation of RCM forcing because of internal variability, but: since you are bias correcting over a reference period, at least the MB model bias (not RMSE) could be assessed when driven by RCMs as well for glaciers with long observation time series. These data would provide a much better estimate of the true uncertainty of the model driven by RCM data for the future. I'm leaving it open to the authors if they want to implement this validation or not - I believe it would make their paper much stronger.

**1.2 Glacier geometry**

The Huss and Farinotti (2012) approach (HF2012), which is to "squeeze" glaciers into elevation bands is an interesting compromise parametrization, simpler than the multiple flowline algorithm followed by OGGM (Maussion et al., 2018) but still allowing for ice flow considerations. It has some advantages (I don't necessarily agree with the ones listed in the paper): it is programmatically more efficient, arguably more elegant (because simple), and it is probably less sensitive to uncertainties in glacier outlines or topography. It also has some disadvantages (mostly, the lost of geometrical information for more complex MB models, and the over simplification of the mass flow along multiple branches).

In an attempt to reproduce the method following the algorithm description by HF2012, I consistently obtain shorter glaciers than provided by the authors (e.g. as shown in Fig. 5). See https://nbviewer.jupyter.org/github/fmaussion/misc/blob/master/simplified_flowline_tests.ipynb for some code and graphics.

I wonder why I can't reproduce the authors' results, and I therefore have a few questions:

- what motivated the choice of 10m for the $\delta z$ elevation bands? This is quite a narrow range and I get better results with larger bands (depending on the underlying

map resolution).

- what do you do when there is no glacier grid point in a 10m band? This happens quite often depending on the underlying map resolution (see graphs).

- do you do any kind of filtering for large slopes? The skewed slope distribution towards high slopes can affect the mean and, together with the missing bands, could explain why I get shorter glaciers.

- do you do apply any smoothing on the resulting band widths and areas? They appear quite noisy in my case (depending on the underlying map resolution).

I'd like to see these questions answered in this manuscript, unless I missed them from either HF2012 or Huss and Hock (2015), in which case I'm happy to be corrected and pointed to the location where the algorithm is described.

Similarly, there are some locations in the current manuscript where I find that the algorithm description is too vague to allow reproducibility (see specific comments below).

**2  Specific comments**

**Abstract L10** : "which the latter" sounds strange. Rephrase?

**Abstract L20** "RCM that is coupled to it" → the RCM is not "coupled" to the GCM (this suggests two-way nesting) ; maybe "nested in", or "driven by" the GCMs? Also revise other occurences in the text.

**P2L13** "the evolution of the glacier" → "glacier evolution"

**P2L21** "moderate" and "extreme" are subjective adjectives → be more precise, e.g. RCP or similar

**Legend Fig 1**  updated version OF Huss and ...

**P3L23**  "we aim at reducing the considerable uncertainties" → I'm yet to be convinced
that increased complexity reduces uncertainty, and I'm not sure your study really
deals with this topic or even actually shows that uncertainties are reduced. It's
okay if you leave this sentence as is, but you don't need this paragraph to justify
your study.

**P4L3**  what is the "local surface slope"? According to HF2012 it is the bin average for
each elevation band. Be more precise in the formulation here (see also general
comment about that).

**P4L7**  "little-known connections" → I don't understand what you mean. Connections
are maybe more complex in a dynamical sense but so are other locations on the
glacier as well. Furthermore, "ignoring" these connections is not making them
less complex, it's just avoiding them. So, I suggest to remove this sentence (see
also general comment)

**P4L9**  trapezoidal sections: how does this go together with the ice thickness inversion?
What cross-sections are used in HF2012? If rectangular (I assume), by using a
trapeze you are either reducing the sections volume or increasing the thickness
$h_0$, i.e. you are not physically consistent between the inversion and the forward
model.

**P4L17**  "close representation of past temperature and precipitation and certain events"
→ Reanalysis datasets also represent weather events well thanks to data assim-
ilation. It's okay to use ENSEMBLES, but you should argue otherwise, maybe
because of uncertainties in quantitative precipitation estimates or the coarse res-
olution of reanalysis data, for example.

**P4L24**  I think this whole justification paragraph is more confusing than helping. I think

it's okay to use an observational dataset for calibration and validation instead of reanalysis, consider shortening this paragraph.

**P4L28** is "chains" the commonly used word for this? I thought that "realisation" or "simulations" would be more appropriate. See also other occurrences in text.

**P5 Eq. (2)** I have several questions here. First, you don't say over which observational period you compute the averages for the monthly bias correction. Is it 1961-1990? The entire observation period? I assume that $\sigma_{obs}$ and $\sigma_{obs}$ is computed for the same reference period as the bias. Then, why choosing a 25-yr period, and not a period of the same length as the reference period? Please also add a sentence as to why you don't apply such a correction for precipitation. I understand that the arithmetics are not so easy for multiplicative bias corrections, but in theory some kind of correction would also be possible (and might be needed by looking at Fig. 02).

**P5L26** "based on a combined criterion weighting both horizontal distance and the difference in area." Can you be more specific here? (reproducible science versus "black box"). How many glaciers have Geodetic and traditional MB observations? Which area does it represent? How many glaciers needed this kind of interpolation?

**P7L25** what kind of numerical solver are you using? It's not an harmless choice, as shown by Jarosch et al 2013.

**P8L10** "Notice that through this approach, the glacier is not assumed to be in steady state at any point in time, but that an artificially modelled steady state is obtained by imposing a MB offset." → I don't understand what you want to say here. I'm also quite confused at the statement "A determines volumes, SMB bias determines the length". Is this based on you own experience, or is there a physical explanation? Finally (and most importantly), why is length used as

convergence criterion instead of area, which is the only variable which is almost perfectly known at the inventory date?

**Model initialisation** needless to say, the iterative initialisation procedure is... unconventional. I'm not asking to change it, because it serves one purpose: find a transient glacier which is consistent with the forward model at a reference date. This is necessary because the ice-thickness inversion model and the forward model in GloGEMFlow are probably not consistent between each other (different MB profiles, different A, different bed shapes). However, I would like to add that I don't really think that this iterative method has much to do with finding an "appropriate" A for each glacier. Let's take the first step as an example: since you drive your model with an SMB such that the present day geometry is in equilibrium, modifying A so that your glacier has to grow will always tend towards lower values of A in order to create a thicker, longer equilibrium glacier in 1990.

**P8L23** this cannot be considered an "independent" validation (see general comment)

**P8L25** "rather than the coupled SMB – ice flow model" → this is a bit of a missed opportunity, because there are chances that the varying geometry actually improves the SMB validation, by taking geometry changes into account which are present in observations but not in the static model.

**Fig 4 Legend** r2 is the "coefficient of determination"

**P8L29** elevation bands and correlation → I agree with Ben Marzeion

**Fig 5** intuitively, I would swap the glacier flow direction so that the distance on model grid (x-axis) is starting from zero at the glacier top. This would also allow to read the length of the glacier directly on the x-axis

**P9L21** how did you compute the surface velocity out of the depth-integrated velocity given by the shallow-ice approximation?

**P10L12** note that other length records are also available for the non-swiss glaciers (WGMS or Leclerq database)

**P10L20** remove "highly significant" and the p-value to read "the correlation is r2 = 0.37 (p < 1e-3)"

**P11L4** unit km2 yr-1

**P12L6** "highest correlation with the maximum glacier elevation" → is this sentence correct?

**Fig 9 Legend** remove the "two" in "two present day"?

**Fig. S3** Consider adding Fig. S3 to the main manuscript.

**P14L10** what do you mean with "ice is more pronounced"?

**P15L12** when the variable IS considered? I'm not sure I fully understood this section.

---

## Author Comment (AC1) · 1 Mar 2019

**Reviewer 1 – Ben Marzeion**

**General comment**

> **[RC1.01]** Zekollari and colleagues present projections for glaciers in the European Alps using a glacier model that explicitly resolves ice dynamics and is forced by GCM projections dynamically downscaled in the framework of CORDEX. The manuscript presents a timely step forward for modeling of glaciers on large regional scales. It is written well and succeeds to make the assumptions and limitations of the model accessible. I am particularly impressed with high quality of the presentation of the results in the Figures. Overall, I have no doubt that this is a valuable contribution to the literature and eventually should be published. However, there are a number of minor points that should be addressed by the authors, as well as three somewhat major issues

We thank the reviewer for taking the time to read our manuscript and for his positive (and very useful!) feedback. All points raised by the reviewer have been addressed and answered, and the manuscript has been updated accordingly.

**Major comments**

> **[RC1.02] Use of CORDEX data**: This is the biggest issue I have: the authors use the projections from CORDEX, present a relatively thourough validation (some comments on this below), but do not at all address a question that seems very obvious to me: from the perspective of modeling glaciers, is there added value in CORDEX? I.e., what difference does it make applying the downscaled data opposed to applying the GCM data directly? I'm aware that because of internal variability, there is not much use in using the CORDEX (or GCM) data during the 2003-2017 validation period in the same way the E-OBS data are used. But there is still some insight to be gained concerning, e.g., temporally accumulated area and volume loss.
>
> Also: through comibination with differen RCMs, the ensemble size of CORDEX is increased - however, there are GCMs not part of CORDEX. Taking the ensemble as a measure of uncertainty, how does the addition of the RCM "axis" affect the ensemble uncertainty? Is is equivalent to taking a larger GCM ensemble, or can insights be gained through the downscaling? Of course, this point is not a weakness of the manuscript. But I see the approach taken by the author as great opportunity to address some questions that are very relevant, since dynamical downscaling is very expensive, and its value is somewhat debated. I think it would be great if the authors took this opportunity.

We thank the reviewer for this suggestion, and agree that it can be instructive to compare the effect of directly forcing the glacier model with GCM outputs rather than RCM outputs (i.e. EURO-CORDEX RCMs, which are driven by a GCM). This is an analysis that we considered including when writing the original manuscript, but which we eventually decided not to include for the sake of conciseness and not to deviate too much from our main message.

We have now reconsidered this, and the updated manuscript now includes new simulations in which the glacier model is directly forced with GCM data. We have however decided to not go into too many details, as this would be beyond the scope of the study and is, probably more suited for a dedicated, more in-depth analysis. Such an in-depth analysis would require a comparison between the GCM and the GCM-RCM simulations themselves, rather than only focusing on the effect this has on the glacier simulations. In the end, the focus of our study is on the glacier model, and not necessarily on the data that drive it.

The focus of our new analyses with GCM forcing is on future glacier evolution, which is also the main period of interest in our paper. For the past, we have now performed an assessment of how the RCM model performs in response to a suggestion made by reviewer #2 (see our response to RC2.05). To stay in the framework of the original manuscript, we decided to only include GCM simulations that are used within EURO-CORDEX (in other

words: we do not consider GCMs that were never used to force EURO-CORDEX RCMs). For this, we considered the three GCMs with the corresponding RCM simulations in the EURO-CORDEX ensemble for a given realisation (CNRM-CM5 r1i1p1, HadGEM2-ES r1i1p1, MPI-ESM-LR r1i1p1).

The results from the GCM-forced simulations support our earlier findings, which indicated that the differences in future glacier evolution are mainly caused by differences in GCM outputs, rather than by differences in the RCMs forced by them (results from section 6.1.4.). In fact, for a given GCM, the spread between EURO-CORDEX simulations forced by the same GCM have a relatively narrow spread. As only few GCM-RCM simulations exist, it is not possible to determine whether these simulations spread evenly around the corresponding GCM simulation. The available simulations, however, do not indicate that any systematic over- or underestimation might occur. As such, we conclude that the 'addition of the RCM axis', as suggested by the reviewer, does not have a significant influence on the results. This suggests that, at least for our setting, using high-resolution RCM simulations forced with a GCM leads to relatively similar results compared to using the GCM forcing directly. As stated above, a more in-depth study would however be needed to further investigate this.

In the updated manuscript, we now introduce these new simulations and present their outcome in section 6.1.4:
*The importance of the GCM-forcing also appears from additional simulations in which the original, low-resolution GCM output was used as model forcing. When comparing these results to the ones obtained by forcing the model with the corresponding EURO-CORDEX GCM-RCM combinations, a similar glacier evolution is obtained (with volume losses vs. present-day typically differing <10%, suppl. mat. Fig. S5). The limited number of GCM-RCM combinations, however, does not allow for a detailed comparison with the GCMs, but does not suggest any systematic over- or underestimation of the corresponding results.*

And we now also mention this in the conclusion:
*Additional simulations where the model is forced with the driving GCM only (i.e. no downscaling with an RCM) confirm the limited effect of the RCM on the modelled future evolution. More in-depth analyses on the effect of using downscaled RCM data vs. GCM data for glacier evolution modelling will be required, but our results suggest that the effect of such a downscaling on simulated glacier evolution is relatively limited – at least for the European Alps.*

A figure in which these results are summarised was added to the supplementary material (suppl. mat. Fig. S5; see following page)

[Figure]

**Fig. S5.** Future evolution of glacier volume when the model is forced directly with GCMs vs. EURO-CORDEX GCM-RCM simulations. Results are shown for the GCMs that have the most corresponding RCM simulations in the EURO-CORDEX ensemble for a given realisation (CNRM-CM5 r1i1p1, HadGEM2-ES r1i1p1, MPI-ESM-LR r1i1p1).

**[RC1.03] Initialization**: Why is glacier length chosen instead of glacier area? It seems to me that area is better constrained than glacier lenght, since the length is to some degree an arbitrary choice resulting from the representation of the flow line.

Also, using the 1961-1990 climatology, was variability preserved? Where the forcing timeseries during that period detrended? As the authors say, there is no assumption about an equilibrium state of the glacier, but an assumption that the glacier woud be able to undergo the transition from an equilibrium state in 1990 to the obsserved transient state in (typically) 2003). Doesn't this correspond to an assumption that the response time of the glaciers is shorter than 13 years? You discuss this in Sect. 6.3, but it would be good to have some arguments presented already in Sect. 3.3.

We agree that the length has a dependence on the methodology used to derive it. However, for the calibration we aim at minimizing the difference between the 'reference' length and the 'calibrated' length. In this sense, considering the length or the area as a measure for this exercise is almost equivalent, as locally the area is the product of the glacier length and width. This is now explicitly mentioned in the updated manuscript and supported with quantitative information:

- We calibrate to the reference glacier length within 1%. The glacier length is thus very closely reproduced, with a standard deviation between the reference and the modelled lengths corresponding to 0.5%.
- For the glacier area, despite not being calibrated to it, the agreement between the reference and the modelled value is almost as good as for the length, with a standard deviation of 0.7%.

Although we agree that the area could have been used instead, we preferred to use the length as a criterion for our calibration, as this ensures that the position of the glacier terminus is correctly reproduced. We now explicitly mention this in the manuscript:

*The glaciers are calibrated to match the length and volume at inventory date within 1% (σ*

*between reference and modelled volume or length of 0.6% or 0.5% of the reference value, respectively). Despite not being calibrated to it, the observed glacier area is also closely reproduced (σ of 0.7%).*

For the period 1961-1990, the variability is not preserved, nor is any detrending applied. This is not needed for our purpose, as this forcing is only used for creating the artificial steady state in 1990. In fact, the constant conditions correspond to the mean SMB over this time period (and an eventual bias), and we have now reformulated this passage to better reflect this (see updated text below)

Our setup does not assume that the glaciers have a response time of less than 13 years. In fact, we rather assume the opposite, i.e. that most glaciers have a response time of more than 13 years, so that the geometry in 2003 still depends on the geometry in 1990. This is important for our specific setup to be used. We argue that this is realistic, as most glaciers were not very far from equilibrium in the late 1980s, and since the typical response time of glaciers, in the order of a few years to decades, suggests that observed evolution in the 1990s and early 2000s was largely determined by their 1990 geometry. We have now emphasized this in section 3.3.:

*The initialisation consists of closely reproducing the glacier geometry at the inventory date. At first, constant climatic conditions are imposed, until a steady state is created, which represents the glacier in 1990 (Fig. 3). These constant climate conditions correspond to the mean SMB under the 1961-1990 climate, to which a SMB perturbation is applied (detailed below). Subsequently, the glacier is forced with E-OBS data, and evolves transiently from 1990 until the glacier-specific inventory date (typically 2003). We opt for a 1990 steady-state glacier, as the glaciers in the European Alps were generally not too far off equilibrium around this period, with SMBs for many glaciers being close to zero (Huss et al., 2010a; WGMS, 2018). By imposing a steady state in 1990, the glacier length at inventory date can be influenced. Methodologically, choosing an initial steady state before 1990 would be problematic, as in this case the glacier geometry would not determine the glacier length at the inventory date anymore, as the period between the steady state and the inventory date exceeds the typical Alpine glacier response time of several years to a few decades (e.g. Oerlemans, 2007; Zekollari and Huybrechts, 2015).*

**[RC1.04] Validation**: The geodetic mass balances used for calibration are most probably not independent of the in situ measurements used for validation. For those glaciers that have geodetic observations that do not temporally overlap with the in situ measurements, it would be good to recalibrate using these, and revalidated using the temporally independent in-situ values.

I'm also wondering about the choice of using geodetic observations for calibration, and in-situ for validation. The opposite would seem like the more natural choice to me, since geodetic MBs can cover longer time periods, and thus allow to include effects from ice dynamics in the total uncertainty. Is there a specific reason not to make the calibration based on in-situ observations, and validate using geodetic MBs?

We agree that in-situ measurements cannot be considered to be entirely independent from the geodetic measurements for cases where both overlap in time (despite relying on different sources and techniques used to derive these values). Following the reviewer's suggestion, the SMB validation has therefore been revised, and we now only consider in-situ measurements that do not overlap in time with the geodetic mass balance measurements. Note that some in-situ measurements refer to glaciers for which no geodetic mass balance exist; these measurements were thus included in the validation as well. These latter measurements are particularly interesting, as they also serve to validate the extrapolation of the geodetic mass balance (cf. comments RC2.02 and RC2.23 by the second reviewer). The text, Figure 4 and its caption have been revised to account for this new validation procedure:

*In order to ensure that the validation procedure is independent from the calibration,*

*validation is only performed with observations that do not temporally overlap with the geodetic mass balances used for calibration (cf. sections 2.3 and 3.1) and for glaciers without geodetic mass balance observations.*

and:

*When only considering SMB measurements on glaciers that have no observed geodetic mass balance (i.e. glaciers for which the geodetic mass balance used to calibrate the model was extrapolated from other, nearby glaciers), the misfit between modelled and observed values increases only little (RMSE = 0.79 m w.e. yr-1; MAD= 0.72 m w.e. yr-1; mean misfit = -0.19 w.e. yr-1), indicating that the method used to extrapolate the geodetic mass balances to unmeasured glaciers performs well.*

Figure 4 was updated to account for this new validation, and so was its caption:
*Fig. 4. Evaluation of modelled SMB against observations from the WGMS (2018) database. All observations are included, except those that do temporally overlap with the geodetic mass balance observations (used for calibration).*

We use the geodetic mass balance for calibration because that is available for many glaciers (for ca. 1500, representing more than 60% of the total glacier area; cf. response to RC2.02). In contrast, only a few glaciers have in-situ measurements. We argue that it is more important to have a good coverage for calibration than for validation, and think that this strategy will be the only one applicable to other regions or the worldwide scale, since the availability of geodetic mass balances massively outgrows the availability of in-situ measurements by far (c.f. works by Brun et al., 2017, Nature Geoscience; Braun et al., 2019, Nature Climate Change). To clarify this, we added the following sentence in section 2.3. (that is where the mass balance data is introduced):
*Note that we prefer using geodetic mass balance over SMB observations for calibration, as we argue that it is more important to have a good coverage for model calibration than for its validation. Furthermore, geodetic mass balances are becoming increasingly available at the regional scale (e.g. Brun et al., 2017; Braun et al., 2019) and outgrow the availability of in-situ measurements, making the adopted strategy applicable to other regions.*

**Minor comments**

**[RC1.05]** P1 L10: ... ice flow processes, _of_ which the latter ...

This was modified.

**[RC1.06]** P1 L10: you could specify it is not included explicitly, since the (simple) parameterizations used are supposed to parameterize its effect.

[note that "it" in the reviewer comment refers to ice dynamics]
We have now specified that this effect was not included *explicitly* in previous studies:
*…, of which the latter is to date not included explicitly in regional glacier projections for the Alps*

**[RC1.07]** P2 L21: specify "glacier model parameters".

Two possible glacier model parameters, which are discussed in the manuscript, have now been added:
*…(e.g. flow parameters and cross-section parameterization).*

**[RC1.08]** P2 L17: either delete "using various methods", or briefly specify.

Deleted.

**[RC1.09]** P2 L18 (and P15 L27): volume/length/area or volume-lenght-area scaling

This was modified as suggested by the reviewer:

*volume/length/area scaling (Marzeion et al., 2012; Radić et al., 2014)*

**[RC1.10]** P2 L21: I suggest RCP 8.5 should not be called "extreme", since the different scenarios are not based on probabilities, nor is there a physical reason to view RCP8.5 above some limit value.

This was now modified to:

*…, and an almost complete disappearance of glaciers under warmer conditions (RCP8.5).*

**[RC1.11]** P3 L2-4: Not sure why glaciers in the Canadian Rockies should not be controlled by topography and local effects. Either specify, or delete; I think there is value in different approaches to similar problems by itself, so I don't see a need for a very strong justification of your study here.

This passage was deleted. The previous sentence now reads:

*In an RGM study for western Canada, Clarke et al. (2015) showed that relative area and volume changes are well represented by such a model, but that large, local present-day differences between observed and modelled glacier geometries can exist after a transient simulation.*

**[RC1.12]** P3 L9: you might also refer to Goosse et al. (2018), which has transient simulations (DOI: 10.5194/cp-14-1119-2018)

This is a good suggestion. We added:

*This model was recently also used by Goosse et al. (2018) to simulate the transient evolution of 71 Alpine glaciers over the past millennium.*

**[RC1.13]** P4 L7: delete "in these little-known connections" (or specify why they are lesser known than the rest of the glacier).

It is not the connections that are little-known, but rather the mass transfer between these connections. We have now reformulated this:

*and potential problems related to solving the little-known mass transfer in these connections.*

**[RC1.14]** P4 L12: I would prefer "Climate data" or "Atmospheric boundary conditions"- but that is maybe a matter of taste.

This section is now entitled: *2.2 Climate Data*

**[RC1.15]** P4 L19: "... has a higher resolution than the reanalysis data used in Huss & Hock (2015, ERA-INTERIM) and goes back further in time."

This was modified in response to RC2.19 and now reads:

*We prefer using an observational dataset compared to a re-analysis product (e.g. ERA-INTERIM, as used in Huss and Hock, 2015), as the former has a higher resolution and goes further back in time.*

**[RC1.16]** P4 L21: rephrase around "several chains used for", since it is not quite clear here

| |
|---|
| what "chains" you are referring to. |
| We now refer to these RCM combinations as *(RCM) simulations* here and throughout the manuscript. |

| |
|---|
| **[RC1.17]** P4 L28 (and throughout the study): perhaps it would be better to call them "model combinations" than "chains" (these "chains" have only two links anyway...). |
| This comment was addressed in our previous response (to RC1.16): the "chains" have been changed to "RCM simulations" throughout the manuscript. |

| |
|---|
| **[RC1.18]** P4 L30: "... a peak-and-decline scenario ..." |
| Modified as suggested. |

| |
|---|
| **[RC1.19]** P4 L32: "Note that while country-specific ..." |
| *while* was added. |

| |
|---|
| **[RC1.20]** P4 L32: "... such as the projections recently released for the CH2018 report, ..." (or something similar - i.e., distinguish between scenarios and projections). |
| This was reformulated to: |
| *Note that while country-specific projections such as the ones recently released with the CH2018 report for Switzerland (CH2018, 2018) exist,…* |

| |
|---|
| **[RC1.21]** P5 L13: because of the difference in resolution between E-OBS and the RCMs, there is probably not a direct equivalent in the grind points. Please specify how you handle this. |
| The trends from the RCMs are imposed on the E-OBS grid to simulate the future SMB. For this, we rely on the nearest grid point. We now mention this in the first sentence of this paragraph: |
| *For modelling the future SMB, debiased RCM trends from the EURO-CORDEX ensemble are imposed on the E-OBS grid based on the nearest corresponding grid cell.* |

| |
|---|
| **[RC1.22]** P6 L13: "closest to the glacier" - there are probably glaciers that are covered by multiple grid cells - are then multiple grid cells taken into account, or are you using the glacier center point? Please specify. |
| In such cases, the glacier centre point is taken into account, which we now also specify: |
| *For every glacier, the model is forced with monthly temperature and precipitation series (section 2.2) from the E-OBS (past) or RCM (future) grid cell closest to the glacier's centre point.* |

| |
|---|
| **[RC1.23]** P6 L15 (and L19): "temperature-index melt model" |
| This was modified for both occurrences in the text: |
| *temperature-index model → temperature-index melt model* |

| |
|---|
| **[RC1.24]** P6 L26-30: While precipitation is the least constrained boundary condition, the degree-day factors are probably the most potent paramteters for tuning. Why is the order chosen like this, and how is the defaul degree-day factor chosen? Please specify. |

We keep the same setup as in GloGEM, in which the precipitation is modified first, and the degree-day factor is potentially altered after that (a third step involves a change in temperature). The reason for this is that local precipitation is often least constrained and most variable in high-mountain areas (e.g. snow redistribution). This is now also mentioned:

*In the first step, overall precipitation is multiplied with a scaling factor varying between 0.8 and 2.0. This initial step focuses on the precipitation, as this is the variable that is expected to be the most poorly reproduced due to resolution issues, spatial variability and local effects (e.g. snow redistribution) (e.g. Jarosch et al., 2012; Hannesdóttir et al., 2015; Huss and Hock, 2015).*

The default degree-day parameter is set to 3 mm d$^{-1}$ K$^{-1}$, in line with the original GloGEM study (Huss and Hock, 2015) and is in agreement with literature values from various studies (Hock, 2003). This is now also mentioned in the text: *…(default value is 3 mm d$^{-1}$ K$^{-1}$; cf. Hock (2003)) and the degree-day factor…*
* * *
**[RC1.25]** P8 L25: It is fine to seperate the validation of the SMB model from the ice dynamics, but there should not be much difference if the dynamics are behaving well. Also, the SMB observations are probably not taking into account the RGI outlines, such that there is some disagreement anyway. Have you tested how much the results differ if you switch on ice dynamics?

It is true that accounting for an evolving geometry while validating the SMB should work well if the glacier evolution model works well. However, we decided to not rely on a dynamically evolving glacier geometry for the SMB validation for the following reasons:
- By relying on a dynamically evolving glacier geometry, only SMB observations after 1990 (starting point of dynamic simulations) can be used for validation, This would reduce the total number of available measurements (note that the number was already reduced in the updated manuscript, to account for the interdependency issue of validation/calibration data raised by the reviewer in RC1.04)
- A related problem is that the changes modelled in glacier area shortly after 1990 are in general too small, as the glaciers needs to evolve away from the steady state (that is also the reason why our validation occurs mainly after the inventory date). Depending on the timing, additional observations may thus have to be removed from the validation dataset.

The sentence has been reformulated in order to reflect this reasoning:
*As the aim is to evaluate the performance of the SMB model (rather than the coupled SMB – ice flow model) and to incorporate as many validation points as possible (which is only possible after 1990 for the dynamic simulations), these calculations are based on the glacier geometry at inventory date.*
* * *
**[RC1.26]** P8 L29-30: I don't agree: this correlations measures the skill of the model to reproduce more positive mass balances at the upper part of the glacier than at the lower parts. A better measure for the SMB model's ability to represent the MB profile would be a comparison of the (temporally averaged) MB profiles, or correlating the deviations from the "climatological" (i.e., mean) profile.

We agree that the original formulation ('*the SMB model distributes the annual SMB relatively well over elevation*') may be perceived as misleading. However, this analysis indicates that the SMB gradient is well reproduced. This has now been reformulated:
*Furthermore, the good agreement between observed and modelled balances for glacier elevation bands (r$^2$ = 0.60; Fig. 4b,d) suggests that, despite not being calibrated to this, the modelled and observed SMB gradient are in reasonably good agreement*
Additional measures, such as the ones proposed by the reviewer, could indeed be interesting to present, but given the limited data available for such analyses (we would

need continuous time series for a meaningfully large set of glaciers), these were not included.
* * *
**[RC1.27]** P10 L20: delete "1x".

This was deleted.
*p-value <1x10$^{-3}$ → p-value <10$^{-3}$*
* * *
**[RC1.28]** P10 L27: delete "In the literature".

By deleting *in the literature*, it gives the impression that this is something we have performed in this study. Although it becomes clearer when reading further that this is not the case, we want to avoid any possible confusion here. We have therefore rephrased this to:
*Glacier area changes in the European Alps have been derived in various studies.*
* * *
**[RC1.29]** P10 L30: "over the period 1973-1998/9".

This was modified.
* * *
**[RC1.30]** P11 L19: "evolution is independent of the scenario".

This has been reformulated to: …*this evolution is independent from the RCP.*
* * *
**[RC1.31]** P12 L6: I don't understand this sentence, please rephrase. Do you mean volume change?

We thank the reviewer for pointing this out, as this was a mistake. The sentence now reads:
*…reveals that under RCP2.6, the relative volume loss has the highest correlation with the elevation range (Fig. 9 and suppl. mat. Table S3; r$^2$ = 0.57).*
* * *
**[RC1.32]** P12 L11: "strong below 3200 m".

Modified.
* * *
**[RC1.33]** P12 L18: delete "may" (you already say it's only under certain model combinations).

*may* was omitted.
* * *
**[RC1.34]** P12 L21: "This is illustrated for Langtaler ..." Sect. 6.1.1: Just out of curiosity, it would be interesting to know about the committed mass loss past 2100.

Langtaler Ferner is projected to lose 89% of its 2017 volume by 2100 under 1988-2017 conditions. After 2100, almost no loss occurs anymore, and slightly less than 10% of its 2017 remains in the end (steady state under 1988-2017 conditions). We now also mention this in the section on the committed loss (6.1.1):
*Under these conditions, the committed loss is particularly strong for small glaciers at lower elevations (e.g. Langtaler Ferner, with a committed volume loss of ca. 90% by 2100)*

**[RC1.35]** P14 L6: correct Section number.

This was corrected:

*…possible to the observed geometry (see section 4.2), which is…*

**[RC1.36]** Appendix: P19 L16: delete "1" at the end of the line.

This was deleted and the sentence was changed to:

*…previous guesses minus 1, e.g. the third guess…*

**[RC1.37]** Table 1: The "committed loss" line is a bit confusing, since the committed loss should not be time-dependent. Perhaps you can call this, e.g., the "realized fraction of committed loss"?

This was indeed somewhat confusing. To clarify this, we now refer to 1988-2017 in the table and in the table caption:

*The evolution under the mean SMB obtained from the 1988-2017 climatic conditions represents the committed loss.*

**[RC1.38]** Generally - but particularly Fig. 9: I'm not a big fan of grids in figures. If there is a need to read numbers/differences from the figures, these numbers should be mentioned on the figure or in the text, and no grid would be necessary; if there is no such need, the grid only adds clutter. Please consider removing grids.

We agree that the grid on the maps in Fig. 9 was not needed and removed this in the updated manuscript. In the other figures we prefer to leave the grids, which we think makes it easier and more convenient to read values in general (without any specific 'highlight' value).

**[RC1.39]** P 27 L3: "relative to 1961-1990" (instead of "with respect to").

This was modified.

**[RC1.40]** P27 L4: is that the mean of all "alpine" EUROCORDEX grid cells, or just the ones that contain glaciers? If the latter, I think it would make more sense to also weight them by glacier area.

We agree with the reviewer that it makes more sense to show the mean temperature and precipitation weighed by the glacier area, and have updated Fig. 2 and its legend accordingly.

**[RC1.41]** Fig. 2: I have a hard time seeing the transparent bands; I suggest deleting them, since that information is already shown in the distribution of the individual (thin) lines.

The transparent bands were omitted in the updated manuscript.

**[RC1.42]** Figs. 5, 6: are there uncertainty estimates available for the observed velocities? If so, it would be good to include them (e.g., just a single error bar so that the observation uncertainty can be compared to the differences with to modeled velocities).

We agree with the reviewer that it is interesting to incorporate error estimates on surface velocities. Unfortunately, uncertainty estimates are often absent for the measurements that we sampled from the literature. Uncertainties in observed velocities are generally <10% (e.g. Berthier and Vincent, 2012; Zekollari et al., 2013; Stocker-Waldhuber et al., 2018). We, thus, argue that the general tendency in this validation will remain unaffected.

---

## Author Comment (AC2) · 1 Mar 2019

**Reviewer 2 – Fabien Maussion**

[RC2.01] The manuscript by Zekollari and colleagues presents new estimations of projected glacier change in the European Alps. It is a well conducted study, the paper is well written and the results are interesting. The inclusion of ice dynamics, the use of CORDEX data instead of coarser GCMs and the large amount of calibration data are the main novel points in this study. It will become the new reference study for future glacier change in the European Alps and as such, it is likely to receive a larger interest from the general public and the media. I have two major concerns that need to be addressed before publication, as well a several specific questions / recommendations.

We thank the reviewer for his generally positive appreciation of the paper. We have addressed the two major points as well as all other specific comments in the revised version of the manuscript.

**General comments - Validation and uncertainty**

I acknowledge the efforts realized to use so many different observational datasets, an exercise only possible in the European Alps. However, I have several issues with the model validation in this study:

[RC2.02] 1. there is no information about how many glaciers (and how much ice area/volume) are simulated without any calibration data. For these glaciers, the geodetic MB is "interpolated" and the effect of this interpolation is not assessed

In total, 1508 glaciers have a geodetic mass balance that can be used for SMB model calibration. By number, this corresponds to 38% of all glaciers. The distribution of glaciers that have a geodetic mass balance, however, is skewed towards larger glaciers, and as a consequence, 60% of the total Alpine glacier area has a geodetic mass balance. We have added this information in the updated manuscript:

*About 1500 glaciers (ca. 38% by number) have a glacier-specific geodetic mass balance observation. Since larger glaciers are overrepresented in this sample, however, this corresponds to about 60% of the total Alpine glacier area.*

What concerns the interpolation of the geodetic mass balance to glaciers without direct observations, the effect is now explicitly addressed (cf. RC1.04). In the updated manuscript, the following passage was added:

*When only considering SMB measurements on glaciers that have no observed geodetic mass balance (i.e. glaciers for which the geodetic mass balance used to calibrate the model was extrapolated from other, nearby glaciers), the misfit between modelled and observed values increases only little (RMSE = 0.79 m w.e. yr$^{-1}$; MAD= 0.72 m w.e. yr$^{-1}$; mean misfit = -0.19 w.e. yr$^{-1}$), indicating that the method used to extrapolate the geodetic mass balances to unmeasured glaciers performs well.*

[RC2.03] 2. there is no indication as to the computation of the uncertainty ranges provided in the glacier changes (e.g. in the abstract). Does it originate from the forcing ensemble? The model RMSE? It is hard make further assessments without this information

It is true that the nature of the uncertainty ranges was not clearly formulated. These uncertainties result from the ensemble of RCM simulations, and this is now explicitly mentioned in the abstract:

*We find that under RCP2.6, the ice loss in the second part of the 21st century is relatively limited and that about one-third (36.8% ± 11.1%, multi-model mean ± 1σ) of the…*

And in the text (when the results are presented for the first time, in section 5):

*Under RCP2.6, in 2100 about 65% of the present-day (2017) volume and area are lost (−63.2±11.1% and −62.1±8.4% respectively, multi-model mean ± 1σ,…*

**[RC2.04]** 3. the validation using observed traditional MB does not make sense to me, because the model has been calibrated on geodetic MB on the same glacier (both data are not exactly the same, but close - see also the comment of Ben Marzeion). If anything, you should use cross-validation here: when assessing the performance of the model on a given glacier, you remove the selected glacier from the calibration dataset, then use this data plus the traditional MB measurements to assess the model. This would also help to address point 1

We agree that the SMB calibration was not fully independent, and have therefore changed the calibration procedure as suggested by the first reviewer (cf. RC1.04). We now make a distinction between different types of validation data, and for validation we only consider SMB observations that do not temporally overlap with geodetic mass balance measurements. Additionally, a comparison between the observed and modelled SMBs is also made for glaciers without any geodetic mass balance observation, which shows that the extrapolation method for geodetic mass balance works well. For more details, we refer to our replies to RC1.04 and RC2.02, where the textual changes are also described.

**[RC2.05]** 4. it is problematic that the effect of the RCM forcing is not assessed at all. The plots all start in 2017, so any sceptic reader could say: "this is all extrapolated without test in the past". I understand the problems behind the validation of RCM forcing because of internal variability, but: since you are bias correcting over a reference period, at least the MB model bias (not RMSE) could be assessed when driven by RCMs as well for glaciers with long observation time series. These data would provide a much better estimate of the true uncertainty of the model driven by RCM data for the future. I'm leaving it open to the authors if they want to implement this validation or not - I believe it would make their paper much stronger.

The idea of adding an analysis of the RCM data in the past is an interesting one. We have incorporated this in part: Rather than performing such an analysis through an SMB validation procedure, we have added a comparison between:
- Past Alpine-wide SMBs obtained by forcing the SMB model with E-OBS data
- Past Alpine-wide SMBs obtained by forcing the SMB model with RCM data

For the latter, we decided to use the historical runs of the EURO-CORDEX models (instead of forcing with ERA-INTERIM). This ensures that the RCM-model skill is assessed, rather than the quality of ERA-INTERIM, and allows for a long comparison period. The comparison shows that, the general tendency and interannual spread in SMB obtained when forcing the SMB model with historical RCM simulations, is comparable to the one when forcing the SMB model with E-OBS data.

This is now described in the manuscript, and a figure was added to the supplementary material (suppl. mat. Fig. S2 in the updated manuscript):

*Finally, sensitivity tests were performed with the SMB model being forced with historical RCM output (instead of E-OBS). The tests indicate that the RCMs, despite not being forced with reanalysis data, are producing general SMB tendencies that are relatively close to those obtained when forcing the model with E-OBS data (similar mean values, see suppl. mat. Fig. S2; similar interannual variability: $\sigma_{SMB,EOBS}$ = 0.66 m w.e. yr$^{-1}$; mean $\sigma_{SMB,RCM}$= 0.58 m w.e. yr$^{-1}$).*

[Figure]

**Fig. S2.** Past specific surface mass balance (SMB) over glaciers in the European Alps, as derived from E-OBS data and from historical RCM simulations. SMB calculations are based on reference geometry at inventory date.

**General comments – Glacier geometry**

**[RC2.05]** The Huss and Farinotti (2012) approach (HF2012), which is to "squeeze" glaciers into elevation bands is an interesting compromise parametrization, simpler than the multiple flowline algorithm followed by OGGM (Maussion et al., 2018) but still allowing for ice flow considerations. It has some advantages (I don't necessarily agree with the ones listed in the paper): it is programmatically more efficient, arguably more elegant (because simple), and it is probably less sensitive to uncertainties in glacier outlines or topography. It also has some disadvantages (mostly, the lost of geometrical information for more complex MB models, and the over simplification of the mass flow along multiple branches).

In an attempt to reproduce the method following the algorithm description by HF2012, I consistently obtain shorter glaciers than provided by the authors (e.g. as shown in Fig. 5). See
https://nbviewer.jupyter.org/github/fmaussion/misc/blob/master/simplified_flowline_tests.ipynb for some code and graphics.

I wonder why I can't reproduce the authors' results, and I therefore have a few questions:

- what motivated the choice of 10m for the δz elevation bands? This is quite a narrow range and I get better results with larger bands (depending on the underlying map resolution)

The reviewer points out some interesting differences between the flowline approach used by OGGM and the one we use. We acknowledge that some of these points were not included in our original submission, and now do so in the reformulated text:

*Subsequently, the glacier geometry is interpolated to a regular, horizontal grid along flow. Through this approach, possible glacier branches and tributaries are not explicitly accounted for, avoiding complications and potential problems related to solving the little-known mass transfer in these connections. As such, this approach is less sensitive to uncertainties in glacier outlines and topography compared to methods in which glacier branches are explicitly accounted for (e.g. Maussion et al., 2018), but may in some cases oversimplify the mass flow for complex glacier geometries (e.g. with several branches).*

In this and the following answers, we argue why the reviewer may have obtained other

glacier lengths compared to us, and explain how we updated the manuscript to further clarify the H&F method applied:
- For the difference in obtained glacier lengths, we refer to our response to RC2.07
- The 10-m elevation bands were chosen to ensure that the method is consistently applicable, also for small glaciers.
* * *
- **[RC2.06]** what do you do when there is no glacier grid point in a 10m band? This happens quite often depending on the underlying map resolution (see graphs).

Having 10-m elevation bands without a glacier grid is not a problem in our case, as the geometric representation with elevation is transformed to a grid with a constant horizontal spacing. This is mentioned in the manuscript:

*The horizontal distance ($\Delta x$) between the elevation bands is determined from the elevation difference ($\Delta y$) and the local surface slope (s):*

*$\Delta x = \Delta y / \tan s$    . (1)*

*Subsequently, the glacier geometry is interpolated to a regular, horizontal grid along flow.*

In case values are missing (i.e. no area and volume for a particular elevation band), these are simply neglected during this interpolation procedure.
* * *
- **[RC2.07]** do you do any kind of filtering for large slopes? The skewed slope distribution towards high slopes can affect the mean and, together with the missing bands, could explain why I get shorter glaciers.

A filtering of the local slopes is performed to get the average slope of elevation bands (that are subsequently used to compute glacier length). This filtering was indeed not described in detail in the original publication (Huss and Farinotti, 2012), thus hampering complete reproducibility. To determine band-average slope, all values below the 5% quantile are discarded, as well as all values above a threshold (typically around the 80 to 90% quantile) determined based on the skewness of the slope distribution function. The approach reduces the effect of very steep cells within an elevation band on average band slope and, hence, glacier length, and has been optimized based on comparisons to flowline glacier length.

This is now formulated after Eq. (1):

*To determine the band-average slope s, all values below the 5% quantile are discarded, as well as all values above a threshold (typically around the 80 to 90% quantile) determined based on the skewness of the slope distribution function.*
* * *
- **[RC2.08]** do you do apply any smoothing on the resulting band widths and areas? They appear quite noisy in my case (depending on the underlying map resolution).

No smoothing is applied of the glacier bands and widths. Despite the fact that the band widths and areas can strongly vary in space (being 'noisy'), they do not lead to any numerical problems when solving the transport equation.
* * *
**[RC2.09]** I'd like to see these questions answered in this manuscript, unless I missed them from either HF2012 or Huss and Hock (2015), in which case I'm happy to be corrected and pointed to the location where the algorithm is described.

Similarly, there are some locations in the current manuscript where I find that the algorithm description is too vague to allow reproducibility (see specific comments below).

By having addressed the comments formulated above and having updated the manuscript accordingly, we hope that the reader will understand the various steps. All specific comments formulated below have been addressed.

**Specific comments**

**[RC2.10]** Abstract L10 : "which the latter" sounds strange. Rephrase?

This has been reformulated to:
*…ice flow processes, of which the latter is to date…*

**[RC2.11]** Abstract L20 "RCM that is coupled to it" → the RCM is not "coupled" to the GCM (this suggests two-way nesting) ; maybe "nested in", or "driven by" the GCMs? Also revise other occurences in the text.

We have now changed this by omitting the 'coupling' part:
*…determined by the driving global climate model (GCM), rather than by the RCM, and…*

This was also updated for other occurrences in the manuscript:
*… an RCM driven by a GCM…* (second last paragraph of introduction)
*… on the driving GCM than the RCM, and …* (last sentence of section 6.1)
*…driving GCM (rather than the RCM), and…* (conclusion)

**[RC2.12]** P2L13 "the evolution of the glacier" → "glacier evolution"

Modified as suggested.

**[RC2.13]** P2L21 "moderate" and "extreme" are subjective adjectives → be more precise, e.g. RCP or similar

This was now modified to:
*These regional and global studies generally suggest a glacier volume loss of about 65-80% between the early 21st century and 2100 under a moderate warming (RCP2.6 and RCP4.5), and an almost complete disappearance of glaciers under warmer conditions (RCP8.5).*

**[RC2.14]** Legend Fig 1 updated version OF Huss and ...

Modified as suggested.

**[RC2.15]** P3L23 "we aim at reducing the considerable uncertainties" → I'm yet to be convinced that increased complexity reduces uncertainty, and I'm not sure your study really deals with this topic or even actually shows that uncertainties are reduced. It's okay if you leave this sentence as is, but you don't need this paragraph to justify your study

We understand the point raised by the reviewer, and now state that we aim at improving future projections and at examining how this could affect global glacier projections (which is indeed more what we do instead of 'reducing the uncertainties'):
*Through novel approaches in terms of (i) climate forcing, (ii) inclusion of ice dynamics, (iii) the use of glacier-specific geodetic mass balance estimates for model calibration, and by (iv) relying on a vast and diverse dataset on ground-truth data for model calibration and validation, we aim at improving future glacier change projections in the European Alps. As a part of our analysis, we explore how the new methods and data utilized could affect other regional and global glacier evolution studies.*

**[RC2.16]** P4L3 what is the "local surface slope"? According to HF2012 it is the bin average

for each elevation band. Be more precise in the formulation here (see also general comment about that).

This comment has been addressed in our reply to RC2.07. In particular, the manuscript was updated in order to clarify how the local surface slope is determined.
* * *
**[RC2.17]** P4L7 "little-known connections" → I don't understand what you mean. Connections are maybe more complex in a dynamical sense but so are other locations on the glacier as well. Furthermore, "ignoring" these connections is not making them less complex, it's just avoiding them. So, I suggest to remove this sentence (see also general comment)

The first reviewer also pointed this out, and we agree that the formulation was not very precise. It is not the connections that are little-known, but rather the mass transfer in these zones. The sentence now reads:

*…complications and potential problems related to solving little-known mass transfer in these connections.*
* * *
**[RC2.18]** P4L9 trapezoidal sections: how does this go together with the ice thickness inversion? What cross-sections are used in HF2012? If rectangular (I assume), by using a trapeze you are either reducing the sections volume or increasing the thickness h0, i.e. you are not physically consistent between the inversion and the forward model.

This is a valuable comment, and we agree that it was not clearly formulated how we treat the different cross section parameterizations. In Huss and Farinotti (2012), a rectangular cross section is used, while we rely on various cross section representations (trapezium with different shapes and rectangular cross section to test for sensitivity to this). In all cases, the cross-section transformation is performed by preserving the area and volume for the particular location. This can lead to slightly different bedrock elevations in this representation, although the differences are in general minor. This is now clarified in the updated manuscript:

*Glacier cross-sections are represented as symmetrical trapezoids. The bedrock elevation is determined in order to ensure local volume and area conservation.*
* * *
**[RC2.19]** P4L17 "close representation of past temperature and precipitation and certain events" → Reanalysis datasets also represent weather events well thanks to data assimilation. It's okay to use ENSEMBLES, but you should argue otherwise, maybe because of uncertainties in quantitative precipitation estimates or the coarse resolution of reanalysis data, for example.

In line with the reviewer's suggestion, this has now been reformulated to:

*This E-OBS product represents past events closely (for example the heat wave of the summer of 2003, Fig. 2b), allowing for detailed comparisons between observed and modelled surface mass balances (section 4.1). We prefer using an observational dataset compared to a re-analysis product (e.g. ERA-INTERIM, as used in Huss and Hock, 2015), as the former has a higher resolution and goes back further in time.*
* * *
**[RC2.20]** P4L24 I think this whole justification paragraph is more confusing than helping. I think it's okay to use an observational dataset for calibration and validation instead of reanalysis, consider shortening this paragraph.

The paragraph was shortened when addressing the reviewer's previous comment (see RC2.19).
* * *
**[RC2.21]** P4L28 is "chains" the commonly used word for this? I thought that "realisation" or

"simulations" would be more appropriate. See also other occurrences in text.

This was now modified to '*simulations*' throughout the text. The wording is classically used in the literature (e.g. Kotlarski et al., 2014, GMD). See also our replies to RC1.16 and RC1.17.

**[RC2.22]** P5 Eq. (2) I have several questions here. First, you don't say over which observational period you compute the averages for the monthly bias correction. Is it 1961-1990? The entire observation period? I assume that σobs and σobs is computed for the same reference period as the bias. Then, why choosing a 25-yr period, and not a period of the same length as the reference period?

Please also add a sentence as to why you don't apply such a correction for precipitation. I understand that the arithmetics are not so easy for multiplicative bias corrections, but in theory some kind of correction would also be possible (and might be needed by looking at Fig. 02).

The bias was evaluated over the longest possible period where both RCM data and E-OBS are available. Some RCMs are available from the 1950s on, while others only start in 1970. The period considered for computing the averages for the monthly bias correction thus ranges from 1970 to 2017, as we now explicitly mention:

*This correction is applied over the period 1970-2017, which is the overlap period for which all RCM simulations and E-OBS data are available.*

A similar correction is not applied for precipitation, as this is a "cumulative" quantity: i.e. monthly differences in variability will not be that relevant at the annual scale (mass budget). Furthermore, variabilities in precipitation do not have a direct effect on the calibrated parameters (as is the case for temperatures via the degree-day factors). This has now been formulated as:

*For precipitation, which enters the SMB calculations as a cumulative quantity, no correction for interannual variability is applied, as the monthly differences in variability are not that relevant at the annual scale. Furthermore, variability in precipitation does not have a direct effect on the calibrated SMB parameters (as is the case for temperatures via the degree-day factors, see section 3.1.).*

**[RC2.23]** P5L26 "based on a combined criterion weighting both horizontal distance and the difference in area." Can you be more specific here? (reproducible science versus "black box"). How many glaciers have Geodetic and traditional MB observations? Which area does it represent? How many glaciers needed this kind of interpolation?

As we stated in our responses to RC1.04 and RC2.02, we now provide more information about the number of glaciers (and the area) that have geodetic mass balance observations. We also added a more elaboration explanation about the procedure that is used to derive the geodetic MB for glaciers without such observations:

*In case no geodetic mass balance observation for the specific glacier is available, an observation from a nearby glacier is chosen. The respective observation is selected based on the two criteria horizontal distance (in km) and relative difference in area (unitless). We multiply the two criteria and consider the minimum as the most suitable glacier to supply a mass balance observation for the unmeasured glacier. The replacement thus represents a nearby glacier that is relatively similar in size. The effect of this approach is evaluated in section 4.1*

**[RC2.24]** P7L25 what kind of numerical solver are you using? It's not an harmless choice, as shown by Jarosch et al 2013.

We agree that it is important to consider stability and mass conservation issues, which are strongly related to the type of numerical solver. Implicit methods allow for using larger time

steps, while explicit methods are intrinsically less stable and need smaller time steps. The latter are however computationally less demanding, and therefore more efficient (see e.g. Schäfer et al., 2007, JGlac). We use a semi-implicit solver. For the calculation of the continuity equation (eq. 6), it relies on an intermediate time step during which the geometry is adapted. We now explicitly mention this in the manuscript.

*The continuity equation is solved using a semi-implicit forward scheme by relying on an intermediate time step (i.e. sub-time step update) in which the geometry is updated.*
* * *
**[RC2.25]** P8L10 "Notice that through this approach, the glacier is not assumed to be in steady state at any point in time, but that an artificially modelled steady state is obtained by imposing a MB offset." → I don't understand what you want to say here. I'm also quite confused at the statement "A determines volumes, SMB bias determines the length". Is this based on you own experience, or is there a physical explanation? Finally (and most importantly), why is length used as convergence criterion instead of area, which is the only variable which is almost perfectly known at the inventory date?

With the sentence formulated on P8L10 in the original manuscript, we want to stress that the steady state that we produce is an artificial one, which we impose for our calibration procedure (to match the geometry at inventory date, see also our response to RC1.03), but we do not assume that the glacier was in steady state with any climatic conditions. In hindsight, we agree that the original sentence was not very clear, and we have decided to omit it altogether.

The second statement ("[parameter] A determines volumes, [the] SMB bias determines the length"), is based on physics, where the rate factor (A) determines the stiffness of the ice, and through this the local ice thickness and thus the ice volume. This is also evident from the equations: if A increases, the local velocity or flux increases, resulting in lower ice thickness. By modifying the SMB bias to create the artificial steady state, the length of the steady state is modified (this is because the SMB needs to be zero when integrated over the glacier). A different steady-state length, in turn, causes the length of the modelled glacier to be modified at inventory as well. We have now clarified this as follows:

*The glacier volume and length at inventory date are matched by calibrating two variables (Fig. 3). The first calibration variable is the deformation-sliding factor A, which mainly determines the volume of the glacier at the inventory date. The reason for this resides in the role that A has on the local velocity/flux, which in turn affects the local ice thickness and thus the ice volume; see Eqs. (4-7). The second calibration variable is an SMB offset in the 1961-1990 climatic conditions used to construct a 1990 steady-state glacier, which mainly determines the length of the steady-state glacier (as the geometry is such that the integrated SMB equals zero). Note that a change in steady-state length causes the glacier length to change at inventory date as well.*

For the comment related to the use of lengths (vs. areas) for calibration, refer to RC1.03.
* * *
**[RC2.26]** Model initialisation needless to say, the iterative initialisation procedure is... unconventional. I'm not asking to change it, because it serves one purpose: find a transient glacier which is consistent with the forward model at a reference date. This is necessary because the ice-thickness inversion model and the forward model in GloGEMFlow are probably not consistent between each other (different MB profiles, different A, different bed shapes).

However, I would like to add that I don't really think that this iterative method has much to do with finding an "appropriate" A for each glacier. Let's take the first step as an example: since you drive your model with an SMB such that the present day geometry is in equilibrium, modifying A so that your glacier has to grow will always tend towards lower values of A in order to create a thicker, longer equilibrium glacier in 1990.

The reviewer suggests that by modifying *A*, the glacier length will be modified. This is rarely the case, as *A* mainly determines the glacier thickness. Through this, it can slightly influence the glacier length (through the SMB – elevation feedback), but this effect is much smaller than the effect that the SMB bias has. We have now clarified this by adapting our manuscript, as we explain in our previous response (reply to RC2.25).

**[RC2.27]** P8L23 this cannot be considered an "independent" validation (see general comment)

See RC1.04: We agree and have now reworked the evaluation procedure by discarding observations that temporally overlap with geodetic mass balances

**[RC2.28]** P8L25 "rather than the coupled SMB – ice flow model" → this is a bit of a missed opportunity, because there are chances that the varying geometry actually improves the SMB validation, by taking geometry changes into account which are present in observations but not in the static model.

See RC1.25: There are several reasons for which we decide not to rely on a dynamically evolving glacier geometry for the SMB validation, and these are now mentioned in the text.

**[RC2.29]** Fig 4 Legend r2 is the "coefficient of determination"

This was modified.

**[RC2.30]** P8L29 elevation bands and correlation → I agree with Ben Marzeion

See RC1.26.

**[RC2.31]** Fig 5 intuitively, I would swap the glacier flow direction so that the distance on model grid (x-axis) is starting from zero at the glacier top. This would also allow to read the length of the glacier directly on the x-axis

Swapping the glacier direction may indeed be an option. But as the figures does not start at zero (it rather shows the distance along the model grid, to ensure a consistency with the figures that will be added as supplementary material), we decided to leave this as it was.

**[RC2.32]** P9L21 how did you compute the surface velocity out of the depth-integrated velocity given by the shallow-ice approximation?

As basal sliding is not treated explicitly in our approach, and given that we assume that the mass transport is defined by the local geometry (SIA), the surface velocities ($\bar{u}$) are equal to the 1.25 x depth-integrated velocities ($u_s$) ($\bar{u}/u_s = 0.8$) (see e.g. Cuffey and Paterson, 2010, p.310). This is now specified:

*In the lower parts, where many glaciers have a distinct tongue, a comparison between observed and modelled surface velocities is possible (surface velocities correspond to 1.25 times the depth-integrated velocities, since we treat basal sliding implicitly, see e.g. Cuffey and Paterson (2010, p.310)).*

**[RC2.33]** P10L12 note that other length records are also available for the non-swiss glaciers (WGMS or Leclerq database)

We now mention the existence of other datasets:

*Note that other length records are also available for non-Swiss glaciers (e.g. Leclercq et al., 2014), but that these were not considered to ensure a consistency in derived length*

*records.*
* * *
**[RC2.34]** P10L20 remove "highly significant" and the p-value to read "the correlation is r2 = 0.37 (p < 1e-3)"

This was modified as suggested.
* * *
**[RC2.35]** P11L4 unit km2 yr-1

Indeed! This was modified.
* * *
**[RC2.36]** P12L6 "highest correlation with the maximum glacier elevation" → is this sentence correct?

This should have read *highest correlation with the glacier elevation range* (cf. Table S3). We thank the reviewer for spotting this!
* * *
**[RC2.37]** Fig 9 Legend remove the "two" in "two present day"?

This was modified.
* * *
**[RC2.38]** Fig. S3 Consider adding Fig. S3 to the main manuscript.

Fig. S3 from the original manuscript was added to the main text and is now Fig. 10. The figure numbering the main text and in the supplementary material has been updated accordingly.
* * *
**[RC2.39]** P14L10 what do you mean with "ice is more pronounced"?

Indeed, this sentence was not clear, as we referred to *ice* as being *more pronounced*, while it should have read *ice loss...more pronounced. We* now modified this accordingly.
* * *
**[RC2.40]** P15L12 when the variable IS considered? I'm not sure I fully understood this section.

We thank the reviewer for pointing this out. This should indeed be 'IS considered' (vs. is not considered in the original manuscript). The text now reads:

*In such an analysis, all independent variables are replaced by dummy variables, which have a value of one when the variable is considered, ….*